# LANGUAGE CONTROL DIFFUSION: EFFICIENTLY SCALING THROUGH SPACE, TIME, AND TASKS

**Edwin Zhang**
Harvard, Founding

**Yujie Lu, Shinda Huang, William Wang**
University of California, Santa Barbara

**Amy Zhang**
UT Austin

## ABSTRACT

Training generalist agents is difficult across several axes, requiring us to deal with high-dimensional inputs (space), long horizons (time), and generalization to novel tasks. Recent advances with architectures have allowed for improved scaling along one or two of these axes, but are still computationally prohibitive to use. In this paper, we propose to address all three axes by leveraging **L**anguage to **C**ontrol **D**iffusion models as a hierarchical planner conditioned on language (LCD). We effectively and efficiently scale diffusion models for planning in extended temporal, state, and task dimensions to tackle long horizon control problems conditioned on natural language instructions, as a step towards generalist agents. Comparing LCD with other state-of-the-art models on the CALVIN language robotics benchmark finds that LCD outperforms other SOTA methods in multi-task success rates, whilst improving inference speed over other comparable diffusion models by 3.3x~15x. We show that LCD can successfully leverage the unique strength of diffusion models to produce coherent long range plans while addressing their weakness in generating low-level details and control.

## 1 INTRODUCTION

It has been a longstanding dream of the AI community to be able to create a household robot that can follow natural language instructions and execute behaviors such as cleaning dishes or organizing the living room (Turing & Haugeland, 1950; Bischoff & Graefe, 1999). Generalist agents such as these are characterized by the ability to plan over long horizons, understand and respond to human feedback, and generalize to new tasks based on that feedback. Language conditioning is an intuitive way to specify tasks, with a built-in structure enabling generalization to new tasks. There have been many recent developments to leverage language for robotics and downstream decision-making and control, capitalizing on the recent rise of powerful reasoning capabilities of large language models (LLMs) (Ahn et al., 2022; Liang et al., 2022; Jiang et al., 2022; Li et al., 2022a).

However, current methods have a few pitfalls. Many existing language-conditioned control methods utilizing LLMs assume access to a high-level discrete action space (e.g. switch on the stove, walk to the kitchen) provided by a lower-level skill oracle (Ahn et al., 2022; Huang et al., 2022; Jiang et al., 2022; Li et al., 2022a). The LLM will typically decompose some high-level language instruction into a set of predefined skills, which are then executed by a control policy or oracle. However, a fixed set of predefined skills may preclude the ability to generalize to novel environments and tasks. Creating a language-conditioned policy that performs effective direct long-horizon planning in the original action space remains an open problem.

One promising candidate that has emerged for direct long-horizon planning is denoising diffusion models. Diffusion models have recently been proposed and successfully applied for low-dimensional, long-horizon planning and offline reinforcement learning (RL) (Janner et al., 2022; Wang et al., 2022b; Chi et al., 2023), beating out several prior Transformer or MLP-based models Chen et al. (2021); Kumar et al. (2020). In particular, Diffuser (Janner et al., 2022) has emerged as an especially promising planner for low-dimensional proprioceptive state spaces with several properties suited for

---

Code and visualizations available at `https://lcd.eddie.win`. Our task is best exemplified by videos available at this URL.

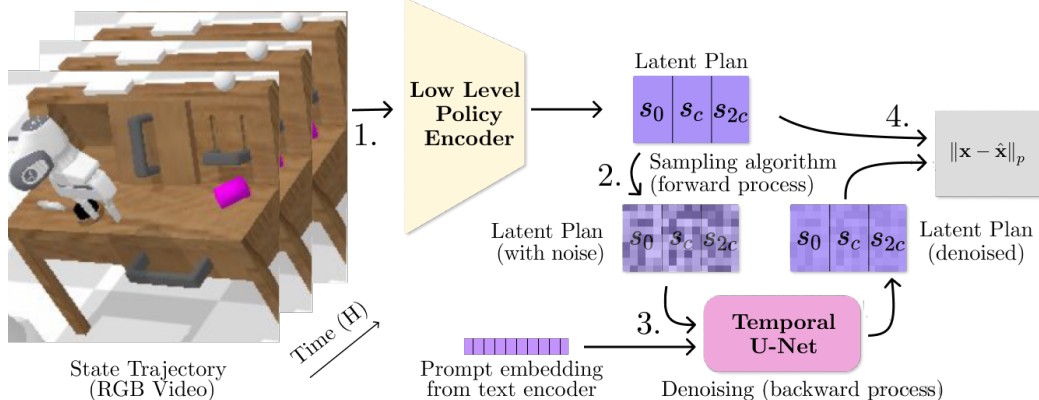

Figure 1: An overview of our high-level policy training pipeline. The frozen low-level policy encoder is used to encode a latent plan, or a subsampled sequence of RGB observations encoded into a lower dimensional latent space (1), which will be used later on as goals for the goal-conditioned low-level policy (LLP). We then noise this latent plan according to a uniformly sampled timestep from the diffusion process' variance schedule (2), and train a Temporal U-Net conditioned on natural language embeddings from a frozen upstream large language model to reverse the noising process (3), effectively learning how to conditionally denoise the latent plan. To train the U-Net, one can simply use the $p$-norm between the predicted latent plan and the ground truth latent plan as the loss (4). We use $p = 1$ in practice following Janner et al. (2022).

language conditioned control: flexibility in task specification, temporal compositionality, and ability to scale to long-horizon settings.

Nevertheless, Diffuser does not work directly out of the box when scaling to pixels. This is perhaps unsurprising, as Diffuser in high-dimensional image space is effectively a text-to-video diffusion model, and even internet-scale video diffusion models have demonstrated only a mediocre understanding of physics and temporal coherence over fine details (Ho et al., 2022a; Singer et al., 2022). This is because the training objective for generative models has no notion of the underlying task, meaning they must model the entire scene with equal weighting. This includes potentially task-irrelevant details that may hinder or even prevent solving the actual control problem (Zhang et al., 2020; Wang et al., 2022a), which can lead to catastrophic failure in control where fine details and precision are essential. Additionally, training a video diffusion model is computationally expensive and may take several days or weeks to train, which leaves such an approach out of reach to most researchers (Rombach et al., 2022). In summary, diffusion models are computationally prohibitive to run on high-dimensional input spaces and also tend to be inaccurate at control in high-dimensional image space.

In this paper, we aim to take a step towards training practical generalist agents. We address all three issues of direct long-horizon planning, non task-specific representation, and computational inefficiency by proposing the usage of a hierarchical diffusion policy.[1] By utilizing a diffusion policy, we can avoid manual specification of fixed skills while keeping flexible long-horizon generation capabilities that other architectures may not have. In addition, we solve both the representation and efficiency issue of a naive application of diffusion by applying the frozen representation of a goal-conditioned policy as a low-level policy (LLP) (see Step 1 in Figure 1, and more details in Appendix C).

In consequence, this hierarchical approach allows us to scale the diffusion model along three orthogonal axes: the **Spatial dimension** through a low-dimensional representation that has been purposely optimized for control, the **Time dimension** through a temporal abstraction enabled by utilizing a goal-conditioned low-level policy (LLP), as the LLP can use goal states several timesteps away from the current state, and the **Task dimension** through language, as the diffusion model acts as a powerful interpreter for plugging any large language model into control. In addition, the entire pipeline is fast and simple to train, as we utilize DDIM (Song et al., 2020a), a temporal abstraction on the horizon, as well as a low-dimensional representation for generation. We can achieve an average of **88.7%** success rate across all tasks on the challenging CALVIN language-conditioned robotics benchmark.

---

[1]To be clear, we are referring to a hierarchical policy where the high-level policy is a diffusion model, not a hierarchical diffusion model that utilizes multiple diffusion models to scale generation (Ho et al., 2022a).

Additionally, we elucidate where diffusion models for text-to-control work well and highlight their limitations.

In summary, our core contributions are: **1)** We propose an effective method for improving diffusion policies' scaling to high-dimensional state spaces, longer time horizons, and more tasks by incorporating language and by scaling to pixel-based control. **2)** We significantly improve both the training and inference time of diffusion policies through DDIM, temporal abstraction, and careful analysis and choice of image encoder. **3)** A successful instantiation of our language control diffusion model that substantially outperforms the state of the art on the challenging CALVIN benchmark.

## 2 BACKGROUND

**Reinforcement Learning.** We formulate the environment as a Markov decision process (MDP) $\mathcal{M} = (\mathcal{S}, \mathcal{A}, \mathcal{R}, \gamma, p)$, with state space $\mathcal{S}$, action space $\mathcal{A}$, reward function $\mathcal{R}$, discount factor $\gamma$, and transition dynamics $p$. At each time step $t$, agents observe a state $s \in \mathcal{S} \subseteq \mathbb{R}^n$, take an action $a \in \mathcal{A} \subseteq \mathbb{R}^m$, and transition to a new state $s'$ with reward $r$ following $s', r \sim p(\cdot, \cdot | s, a)$. The goal of RL is then to learn either a deterministic policy $\pi : \mathcal{S} \mapsto \mathcal{A}$ or a stochastic policy $a \sim \pi(\cdot | s)$ that maximises the policy objective, $J(\pi) = \mathbb{E}_{a \sim \pi; r, s' \sim p} \sum_{t=0}^{\infty} \gamma^t r_t$.

**Language Conditioned RL.** We consider a language-conditioned RL setting where we assume that the true reward function $\mathcal{R}$ is unknown, and must be inferred from a natural language instruction $L \in \mathscr{L}$. Formally, let $\mathcal{F}$ be the function space of $\mathcal{R}$. Then the goal becomes learning an operator from the language instruction to a reward function $\psi : \mathscr{L} \mapsto \mathcal{F}$, and maximizing the policy objective conditioned on the reward function $\psi(L)$: $J(\pi(\cdot \mid s, \mathcal{R})) = \mathbb{E}_{a \sim \pi; r, s' \sim p} \sum_{t=0}^{\infty} \gamma^t r_t$. This formulation can be seen as a contextual MDP (Hallak et al., 2015), where language is seen as a context variable that affects reward but not dynamics. We assume access to a prior collected dataset $\mathcal{D}$ of $N$ annotated trajectories $\tau_i = \langle (s_0, a_0, ... s_T), L_i \rangle$. The language conditioned policy $\pi_\beta$, or the behavior policy, is defined to be the policy that generates the aforementioned dataset. This general setting can be further restricted to prohibit environment interaction, which recovers offline RL or imitation learning (IL). In this paper, we assume access to a dataset of expert trajectories, such that $\pi_\beta = $ optimal policy $\pi^\star$. Although this may seem like a strong assumption, the setting still poses a challenging learning problem as many unseen states will be encountered during evaluation and must be generalized to. Several prior methods have failed in this setting (De Haan et al., 2019; Ross et al., 2011).

**Goal Conditioned Imitation Learning and Hierarchical RL.** Goal-conditioned reinforcement learning (RL) is a subfield of RL that focuses on learning policies that can achieve specific goals or objectives, rather than simply maximizing the cumulative reward. This approach has been extensively studied in various forms in the literature (Li et al., 2022a; Arora & Doshi, 2018; Sodhani et al., 2021; Harrison et al., 2017; Schaul et al., 2015; Rosete-Beas et al., 2023), although we focus on the hierarchical approach (Sutton et al., 1999). Following (Nachum et al., 2018), we formulate the framework in a two-level manner where a high level policy $\pi_{\text{hi}}(\tilde{a}|s)$ samples a goal state $g = \tilde{a}$ every $c$ time steps. However, whilst most prior work only considers generating a single goal forward, we can leverage the unique capabilities of diffusion models to do long-horizon planning several goal steps forward (Janner et al., 2022). We refer to $c$ as the temporal stride. A deterministic low level policy (LLP) $a = \pi_{\text{lo}}(s_t, g_t)$ then attempts to reach this state, which can be trained through hindsight relabelling (Andrychowicz et al., 2017) the offline dataset $\mathcal{D}$. In general this LLP will minimize action reconstruction error, or $\mathcal{L}_{\text{BC}} = \mathbb{E}_{s_t, a_t, s_{t+c} \sim \mathcal{D}} \left[ \|a_t - \pi_{\text{lo}}(s_t, g = s_{t+c})\|^2 \right]$. For our specific training objective, please see Appendix C

## 3 THE LANGUAGE CONTROL DIFFUSION (LCD) FRAMEWORK

In this section, we develop the Language Control Diffusion framework to improve generalization of current language conditioned policies by avoiding the usage of a predefined low level skill oracle. Furthermore, we enable scaling to longer horizons and sidestep the computational prohibitiveness of training diffusion models for control directly from high-dimensional pixels. We start by deriving the high-level diffusion policy training objective, and go on to give a theoretical analysis on the suboptimality bounds of our approach by making the mild assumption of Lipschitz transition dynamics. We

then solidify this theoretical framework into a practical algorithm by describing the implementation details of both the high-level and low-level policy. Here we also detail the key components of our method that most heavily affect training and inference efficiency, and the specific model architectures used in our implementation for the CALVIN benchmark (Mees et al., 2022b).

## 3.1 HIERARCHICAL DIFFUSION POLICIES

**High-level Diffusion Policy Objective.** We first describe our problem formulation and framework in detail. Since we assume that our dataset is optimal, the policy objective reduces to imitation learning:

$$\min_{\pi} \mathbb{E}_{s,\mathcal{R}\sim\mathcal{D}} \left[ D_{\mathrm{KL}} \left( \pi_{\beta}(\cdot \mid s, \mathcal{R}), \pi(\cdot \mid s, \mathcal{R}) \right) \right]. \tag{1}$$

As we are able to uniquely tackle the problem from a planning perspective due to the empirical generative capabilities of diffusion models, we define a state trajectory generator as $\mathcal{P}$ and switch the atomic object from actions to state trajectories $\boldsymbol{\tau} = (s_0, s_1, ..., s_T) \sim \mathcal{P}(\cdot|\mathcal{R})$. Thus we aim to minimize the following KL:

$$\min_{\mathcal{P}} D_{\mathrm{KL}} \left( \mathcal{P}_{\beta}(\boldsymbol{\tau} \mid \mathcal{R}), \mathcal{P}(\boldsymbol{\tau} \mid \mathcal{R}) \right) = \min_{\mathcal{P}} \mathbb{E}_{\boldsymbol{\tau}, \mathcal{R}\sim\mathcal{D}} \left[ \log \mathcal{P}_{\beta}(\boldsymbol{\tau} \mid \mathcal{R}) - \log \mathcal{P}(\boldsymbol{\tau} \mid \mathcal{R}) \right]. \tag{2}$$

This can be reformulated into the following diffusion training objective:

$$\min_{\theta} \mathbb{E}_{\boldsymbol{\tau}_0, \boldsymbol{\epsilon}} [\|\boldsymbol{\epsilon}_t - \boldsymbol{\epsilon}_{\theta}(\sqrt{\bar{\alpha}_t}\boldsymbol{\tau}_0 + \sqrt{1 - \bar{\alpha}_t}\boldsymbol{\epsilon}_t, t)\|^2]. \tag{3}$$

Here $t$ refers to a uniformly sampled diffusion process timestep, $\boldsymbol{\epsilon}_t$ refers to noise sampled from $\mathcal{N}(\mathbf{0}, \mathbf{I})$, $\boldsymbol{\tau}_0$ refers to the original denoised trajectory $\boldsymbol{\tau}$, $\alpha_t$ denotes a diffusion variance schedule, and $\bar{\alpha}_t := \prod_{s=1}^{t} \alpha_s$. We refer to Appendix F for a more detailed derivation of our objective.

**Near Optimality Guarantees.** To theoretically justify our approach and show that we can safely do diffusion in the LLP encoder latent space without loss of optimality up to some error $\epsilon$, we prove that a near optimal policy is always recoverable with the LLP policy encoder under some mild conditions: that our low-level policy has closely imitated our expert dataset and that the transition dynamics are Lipschitz smooth. Lipschitz transition dynamics is a fairly mild assumption to make (Asadi et al., 2018; Khalil, 2008). Several continuous control environments will exhibit a Lipschitz transition function, as well as many robotics domains. If an environment is not Lipschitz, it can be argued to be in some sense unsafe (Berkenkamp et al., 2017). Moreover, one could then impose additional action constraints on such an environment to make it Lipschitz again.

In this paper, we consider high dimensional control from pixels, and thus factorize our low level policy formulation into $\pi_{\mathrm{lo}}(s_t, g_t) := \phi(\mathcal{E}(s_t), g_t)$ where $z := \mathcal{E}(s_t)$ defines an encoder function that maps $s$ into a latent representation $z$ and $\phi$ translates the representation $z$ into a distribution over actions $a$ given a goal $g$. Note that $\phi$ is an arbitrary function, and can be made as simple as a matrix or as complex as another hierarchical model. Let $\pi_{\mathrm{lo}}(s) := \phi \circ \mathcal{E}(s)$ be a deterministic low-level policy. Here, the $\circ$ symbol refers to function composition. Similar to Nachum et al. (2019), we can then define the *sub-optimality* of the action and state space induced by $\tilde{\mathcal{A}} = \tilde{\mathcal{S}} = \mathcal{E}(s)$ to be the difference between the best possible high-level policy $\pi_{\mathrm{hi}}^* = \mathrm{argmax}_{\pi \in \Pi} J(\pi_{\mathrm{hi}})$ within this new abstract latent space and the best policy in general:

$$\mathrm{SubOpt}(\mathcal{E}, \phi) = \sup_{s \in S} V^{\pi^*}(s) - V^{\pi_{\mathrm{hi}}^*}(s). \tag{4}$$

Intuitively, this captures how much potential performance we stand to lose by abstracting our state and action space.

**Proposition 3.1.** *If the transition function $p(s'|s, a)$ is Lipschitz continuous with constant $K_f$ and* $\sup_{s \in S, a \in A} |\pi_{\mathrm{lo}}(s) - a^*| \leq \epsilon$, *then*

$$\mathrm{SubOpt}(\mathcal{E}, \phi) \leq \frac{2\gamma}{(1 - \gamma)^2} R_{max} K_f \mathrm{dom}(P(s'|s, a))\epsilon. \tag{5}$$

Here, $\mathrm{dom}(P(s'|s, a))$ is the cardinality of the transition function. Crucially, this shows that the suboptimality of our policy is bounded by $\epsilon$ and that our framework can recover a near-optimal

---

**Algorithm 1** Hierarchical Diffusion Policy Training

---

**Input**: baseline goal-conditioned policy $\pi_{\text{lo}} := \phi(\mathcal{E}(s_t), g_t)$, diffusion variance schedule $\alpha_t$, temporal stride $c$, language model $\rho$

**Output**: trained hierarchical policy $\pi(a_t|s_t) := \pi_{\text{lo}}(a_t|s_t, \pi_{\text{hi}}(g_t|s_t))$, where $g_t$ is sampled every $c$ time steps from $\pi_{\text{hi}}$ as the first state in latent plan $\boldsymbol{\tau}^c$.

1: Collect dataset $\mathcal{D}_{\text{onpolicy}}$ by rolling out trajectories $\boldsymbol{\tau} \sim \pi_{\text{lo}}, \rho$
2: Instantiate $\pi_{\text{hi}}$ as diffusion model $\epsilon_\theta(\boldsymbol{\tau}_{\text{noisy}}, t, \rho(L))$
3: **repeat**
4:     Sample mini-batch $(\boldsymbol{\tau}, L) = B$ from $\mathcal{D}_{\text{onpolicy}}$.
5:     Subsample latent plan $\boldsymbol{\tau}^c = (\mathcal{E}(s_0), \mathcal{E}(s_c), \mathcal{E}(s_{2c}), ..., \mathcal{E}(s_T))$.
6:     Sample diffusion step $t \sim \text{Uniform}(\{1, ..., T\})$, noise $\epsilon \sim \mathcal{N}(\mathbf{0}, \mathbf{I})$
7:     Update high-level policy $\pi_{\text{hi}}$ with gradient $-\nabla_\theta \|\boldsymbol{\epsilon} - \boldsymbol{\epsilon}_\theta(\sqrt{\bar{\alpha}_t}\boldsymbol{\tau}^c + \sqrt{1 - \bar{\alpha}_t}\epsilon, t, \rho(L))\|^2$
8: **until** converged

---

policy with fairly mild assumptions on the environment. We give a detailed proof in Appendix G. The beauty of this result lies in the fact that due to the nature of the LLP's training objective of minimizing reconstruction error, we are directly optimizing for Proposition 3.1's second assumption $\sup_{s \in S, a \in A} |\pi_{\text{lo}}(s) - a^*| \leq \epsilon$. Moreover, if this training objective converges, it achieves an approximate $\pi^*$-irrelevance abstraction (Li et al., 2006). Therefore, utilizing the LLP encoder space for diffusion is guaranteed to be near-optimal as long as the LLP validation loss is successfully minimized. However, there still exists gaps between the theory and practice, which we detail in Appendix G.

## 3.2 PRACTICAL INSTANTIATION

We now describe our practical implementation and algorithm instantiation in this section. We first give an overview of our high-level policy instantiation before specifying the implementation of our low-level policy. Finally, we describe our model architecture in detail.

**High-Level Policy.** In Algorithm 1 we clarify the technical details of our high-level diffusion policy training. Before this point, we assume that a low-level policy (LLP) has been trained. Whilst our framework is flexible enough to use any LLP trained arbitrarily as long as it belongs to the factorized encoder family described in Section 3.1, in our practical implementation we first train our LLP through hindsight relabelling. As our choice of diffusion policy allows tractably modeling long-horizon plans as opposed to other architectures, we proceed to prepare a dataset of long-horizon latent plans. We take the pretrained encoder representation and use it to compress the entire offline dataset of states. This is useful from a practical perspective as it caches the computation of encoding the state, often allowing for 100-200x less memory usage at train time. This enables significantly more gradient updates per second and batch sizes, and is largely why our method is drastically more efficient than prior work utilizing diffusion for control. After caching the dataset with the pretrained encoder, we then induce a temporal abstraction by subsampling trajectories from the dataset, taking every $c^{\text{th}}$ state in line 5 of Algorithm 1, where $c$ refers to the temporal stride. This further increases efficiency and enables flexible restructuring of computational load between the high-level and low-level policy, simply by changing $c$. Our preparation of the dataset of latent plans is now complete. These plans can now be well modeled with the typical diffusion training algorithm, detailed in lines 6-7. During inference, the LLP will take as input the first goal state in the latent plan and the current environment state, and output actions. After $c$ timesteps, we resample a new latent plan from the HLP and repeat this process until either the task has been finished or a fixed timeout (360 timesteps).

**Low-Level Policy.** We adopt the HULC architecture (Mees et al., 2022a) for our low-level policy, and the state encoder as the visual encoder within the HULC policy. In HULC, the authors propose an improved version of the hierarchical Multi-Context Imitation Learning (MCIL). HULC utilizes hierarchy by generating global discrete latent plans and learning local policies that are conditioned on this global plan. Their method focuses on several small components for increasing the effectiveness of text-to-control policies, including the architectures used to encode sequences in relabeled imitation learning, the alignment of language and visual representations, and data

augmentation and optimization techniques. We did not choose to use a simpler flat MLP as we aimed to implement the strongest possible model. For a more detailed explanation, please refer to Appendix C.

**Model Architecture.** We adopt the T5-XXL model (Raffel et al., 2020) as our textual encoder, which contains 11B parameters and outputs 4096 dimensional embeddings. T5-XXL has similarly found success in the text to image diffusion model Imagen (Ho et al., 2022a). We utilize a temporal U-Net (Janner et al., 2022), which performs 1D convolutions across the time dimension of the latent plan rather than the 2D convolution typical in text-to-image generation. This is motivated by our desire to preserve equivariance along the time dimension but not the state-action dimension. In addition, we modify the architecture in Janner et al. (2022) by adding conditioning via cross attention in a fashion that resembles the latent diffusion model (Rombach et al., 2022). Finally, we use DDIM (Song et al., 2020a) during inference for increased computational efficiency and faster planning. DDIM uses strided sampling and is able to capture nearly the same level of fidelity as typical DDPM sampling (Ho et al., 2020) with an order of magnitude speedup. For rolling out the latent plans generated by the denoiser, we resample a new sequence of goals $g$ with temporal stride or frequency $c$, until either the task is completed successfully or the maximum horizon length is reached. Our low-level policy takes over control between samples, with goals generated by the high-level policy as input.

## 4 EXPERIMENTS

In our experiments we aim to answer the following questions. In subsection 4.2 and subsection 4.3 we answer 1) Does a diffusion-based approach perform well for language-conditioned RL? In subsection 4.4 we answer 2) How much efficiency is gained by planning in a latent space? Finally, in subsection 4.5 and subsection 4.7 we answer 3) is a hierarchical instantiation of the diffusion model really necessary, and what trade-offs are there to consider?

### 4.1 EXPERIMENTAL SETUP

**Dataset and Metric.** We evaluate on the CALVIN, and CLEVR-Robot benchmark (Mees et al., 2022b; Jiang et al., 2019), both challenging multi-task, long-horizon benchmarks. After pretraining our low-level policy on all data, we freeze the policy encoder and train the diffusion temporal U-Net using the frozen encoder. For more details on our training dataset and benchmarks, please refer to Appendix H. We roll out all of our evaluated policies for 1000 trajectories on all 34 tasks, and all comparisons are evaluated in their official repository[2].

**Baselines.** We compare against the prior state of the art model Skill Prior Imitation Learning (SPIL) (Zhou et al., 2023), which proposes using skill priors to enhance generalization. As introduced in Section 3.2, we also compare against HULC and MCIL (Mees et al., 2022a; Lynch & Sermanet, 2020b). HULC provides a strong baseline as it is also a hierarchical method. MCIL (Multicontext Imitation Learning) uses a sequential CVAE to predict next actions from image or language-based goals, by modeling reusable latent sequences of states. GCBC (Goal-conditioned Behavior Cloning) simply performs behavior cloning without explicitly modeling any latent variables like MCIL, and represents the performance of using the simplest low-level policy. Finally, Diffuser generates a trajectory by reversing the diffusion process. Diffuser is most comparable to our method. However, it utilizes no hierarchy or task-specific representation. In our benchmark, Diffuser-1D diffuses in the latent space of a VAE trained from scratch on the CALVIN dataset, whilst Diffuser-2D diffuses in the latent space of Stable Diffusion, a large internet-scale pretrained VAE (Rombach et al., 2022). We also attempted to train Diffuser directly on the image space of CALVIN, but this model failed to converge on a 8x 2080 Ti server (we trained for two weeks).

For the CLEVR robot environment, We consider the following representative baselines: a Goal-Conditioned Behavior Cloning Transformer, which is trained end to end by predicting the next action given the current observation and language instruction. We also consider using a Hierarchical Transformer, which is trained using the same two-stage training method as our method. The Hierarchical

---

[2]https://github.com/lukashermann/hulc/tree/fb14d5461ae54f919d52c0c30131b38f806ef8db

Table 1: **HULC Benchmark Results**. We compare success rates between our diffusion model and prior benchmarks on multitask long-horizon control (MT-LHC) for 34 disparate tasks. We report the mean and standard deviation across 3 seeds for our method with each seed evaluating for 1000 episodes. We bold the highest performing model in each benchmark category. For hyperparameters please see Appendix D.

| Horizon | GCBC | MCIL | HULC | SPIL | Diffuser-1D | Diffuser-2D | Ours |
|---|---|---|---|---|---|---|---|
| One | $64.7 \pm 4.0$ | $76.4 \pm 1.5$ | $82.6 \pm 2.6$ | $84.6 \pm 0.6$ | $47.3 \pm 2.5$ | $37.4 \pm 3.2$ | $\mathbf{88.7 \pm 1.5}$ |
| Two | $28.4 \pm 6.2$ | $48.8 \pm 4.1$ | $64.6 \pm 2.7$ | $65.1 \pm 1.3$ | $18.8 \pm 1.8$ | $9.3 \pm 1.3$ | $\mathbf{69.9 \pm 2.8}$ |
| Three | $12.2 \pm 4.1$ | $30.1 \pm 4.5$ | $47.9 \pm 3.2$ | $50.8 \pm 0.4$ | $5.9 \pm 0.4$ | $1.3 \pm 0.2$ | $\mathbf{54.5 \pm 5.0}$ |
| Four | $4.9 \pm 2.0$ | $18.1 \pm 3.0$ | $36.4 \pm 2.4$ | $38.0 \pm 0.6$ | $2.0 \pm 0.5$ | $0.2 \pm 0.0$ | $\mathbf{42.7 \pm 5.2}$ |
| Five | $1.3 \pm 0.9$ | $9.3 \pm 3.5$ | $26.5 \pm 1.9$ | $28.6 \pm 0.3$ | $0.5 \pm 0.0$ | $0.07 \pm 0.09$ | $\mathbf{32.2 \pm 5.2}$ |
| Avg horizon len | $1.11 \pm 0.3$ | $1.82 \pm 0.2$ | $2.57 \pm 0.12$ | $2.67 \pm 0.01$ | $0.74 \pm 0.03$ | $0.48 \pm 0.09$ | $\mathbf{2.88 \pm 0.19}$ |

Transformer is equivalent to LCD (our method), other than the usage of a transformer rather than a diffusion model for the high-level policy.

We focus our comparisons for CALVIN on the current strongest models on the CALVIN benchmark in our setting of general policy learning (SPIL and HULC). To ensure a fair comparison, we rigorously follow the CALVIN benchmark's evaluation procedure and build directly from their codebase. SPIL, MCIL, and GCBC results are taken from Zhou et al. (2023) and Mees et al. (2022a), whilst HULC results are reproduced from the original repository. Diffuser was retrained by following the original author's implementation.

## 4.2 LCD Performs Well for Language-Conditioned RL

Table 2: **CLEVR-Robot Results.** Success rates between our diffusion model and a GCBC transformer and Hierarchical Transformer (HT) on 80 disparate tasks. We report the mean and standard deviation across 3 seeds for our method with each seed evaluating for 100 episodes.

| | GCBC | HT | Ours |
|---|---|---|---|
| Seen | $45.7 \pm 2.5$ | $38.0 \pm 2.2$ | $\mathbf{53.7 \pm 1.7}$ |
| Unseen | $46.0 \pm 2.9$ | $36.7 \pm 1.7$ | $\mathbf{54.0 \pm 5.3}$ |
| Noisy | $42.7 \pm 1.7$ | $33.3 \pm 1.2$ | $\mathbf{48.0 \pm 4.5}$ |

As shown in Table 1, we significantly outperform prior methods on the multi-task long-horizon control (MT-LHC) benchmark on every metric, and improve on the strongest prior model's average performance on horizon length one tasks by $4.1\%$, and average horizon length by $0.21$. In addition, we outperform the next best diffusion model by $41.4\%$ on length one tasks and by $2.14$ on average horizon length. Here, average horizon length refers to the average amount of tasks that a given model was able to finish. For example, our model's average horizon length of $2.88$ means it was on average able to finish $2.88$ tasks.

Results in Table 2 show that LCD surpasses the GCBC Transformer and the Hierarchical Transformer in Seen, Unseen, and Noisy settings. In "Seen," LCD achieves a **53.7 ± 1.7** SR, confirming its capacity to perform tasks similar to its training. "Unseen" tests generalization to new NL instructions, with LCD leading at **54.0 ± 5.3** SR, which evidences its ability to parse and act on unfamiliar language commands. In "Noisy" conditions with perturbed language input, LCD attains **48.0 ± 4.5** SR, displaying resilience to language input errors. This answers our first research question on whether a diffusion-based approach can perform well for language-conditioned RL.

## 4.3 LCD can Generalize to New Tasks

We test the task generalization of LCD on a collection of five held out tasks in Table 3. This benchmark is difficult as it requires zero-shot generalization to a new task, which requires generalization across language and states. We find that LCD can successfully generalize, and is able to compose several disparate concepts from the training dataset together such as the color of the blocks, verbs such as lift and push, and object positions and state.

Table 3: Task generalization of LCD on a collection of five held out tasks. We test with 3 seeds and report the mean and std, evaluating on 20 rollouts per task for a total of 100 evaluations.

| Task | Diffuser-1D | Ours |
|---|---|---|
| Lift Pink Block Table | $31.67 \pm 10.27$ | $\mathbf{55.00 \pm 16.33}$ |
| Lift Red Block Slider | $13.35 \pm 10.25$ | $\mathbf{88.33 \pm 8.50}$ |
| Push Red Block Left | $1.64 \pm 2.35$ | $\mathbf{35.00 \pm 7.07}$ |
| Push Into Drawer | $3.34 \pm 4.71$ | $\mathbf{90.00 \pm 10.80}$ |
| Rotate Blue Block Right | $5.00 \pm 4.10$ | $\mathbf{36.67 \pm 14.34}$ |
| Avg SR | $12.67 \pm 3.56$ | $\mathbf{61.00 \pm 7.79}$ |

## 4.4 Inference Time is 3.3x to 15x Faster through Hierarchy

Table 4: Wall clock times for training. Latent dims denotes the size of the latent space that we perform the diffusion generation in. We compare against the same two variants of Diffuser as presented in Table 1.

|  | HULC | SPIL | Diffuser-1D | Diffuser-2D | Ours (HLP only) | Ours (full) |
|---|---|---|---|---|---|---|
| Training (hrs) ($\downarrow$ Lower is Better) | 82 | 86 | 20.8 | 49.2 | **13.3** | 95.3 |
| Inference time (sec) ($\downarrow$) | **0.005** | **0.005** | 1.11 | 5.02 | 0.333 | 0.336 |
| Avg $\nabla$ updates/sec ($\uparrow$) | .5 | .5 | 4 | 2.1 | **6.25** | **6.25** |
| Model size ($\downarrow$) | 47.1M | 54.8M | 74.7M | 125.5M | **20.1M** | 67.8M |
| Latent dims ($\downarrow$) | N/A | N/A | 256 | 1024 | **32** | **32** |

Through the usage of DDIM, temporal abstraction, and low-dimensional generation, we find in Table 4 that LCD is significantly faster during rollout and training than Diffuser. In addition, our method is significantly faster to train than HULC, albeit slower during rollout. This is to be expected, as HULC is not a diffusion model and only requires a single forward pass for generation. However, when comparing to other diffusion models we find that our method is 3.3x-15x faster during inference and 1.5x-3.7x faster during training, answering our second research question regarding how much efficiency is gained by planning in a latent space. All numbers are taken from our own experiments and server for reproducibility and fairness, including baselines.

## 4.5 Is Diffusion Really Necessary?

Table 5: Ablation of our method by comparing against a simple residual eight layer MLP with 2048 hidden dimensions (25.5M total parameters) and a four layer Transformer with 4096 hidden dimensions (1.5M total parameters) as high level policy. We use the same methodology as in Table 1, and report the mean and standard deviation across 3 seeds for our method with each seed evaluated for 1000 episodes. For hyperparameters and more details please see Appendix E

| Task | MLP | Transformer | Ours |
|---|---|---|---|
| One | $86.6 \pm 2.3$ | $85.4 \pm 0.5$ | $\mathbf{88.7 \pm 1.5}$ |
| Two | $64.1 \pm 0.1$ | $60.9 \pm 0.5$ | $\mathbf{69.9 \pm 2.8}$ |
| Three | $48.1 \pm 8.3$ | $42.6 \pm 1.9$ | $\mathbf{54.5 \pm 5.0}$ |
| Four | $35.7 \pm 8.8$ | $29.8 \pm 2.5$ | $\mathbf{42.7 \pm 5.2}$ |
| Five | $25.5 \pm 7.6$ | $20.0 \pm 1.8$ | $\mathbf{32.2 \pm 5.2}$ |
| Avg SR | $2.60 \pm 0.33$ | $2.36 \pm 0.08$ | $\mathbf{2.88 \pm 0.19}$ |

In order to analyze whether diffusion planning actually improves HULC or if the gain comes just from the usage of a high-level policy and to answer our third research question, we perform an ablation study in Table 5 by using a simple MLP and Transformer as a high-level policy, which receives the ground truth validation language embeddings as well as the current state, and attempts to predict the next goal state. An example of the ground truth language is "Go push the red block right", which is never explicitly encountered during training. Instead, similar prompts of the same task are given, such as "slide right the red block". We find that even given the ground truth validation language embeddings (the MLP and Transformer do not need to generalize across language), these ablations significantly underperforms LCD, and slightly underperforms HULC. This suggests that our gains in performance over HULC truly do come from better modeling of the goal space by diffusion.

## 4.6 Robustness to Hyperparameters

In **??** we show that our diffusion method is robust to several hyperparameters including a frame offset $o$ and the hidden dimensions of the model. Frame offset $o$ controls the spacing between goal states in the sampled latent plan, which affects the temporal resolution of the representations. We consider augmenting our dataset by sampling a $d \sim \text{Unif}\{-o, o\}$ for goal state $s_{c+d}$ rather than just $s_c$, potentially improving generalization. However, we find that this effect is more or less negligible. Finally, we find our method is also robust with respect to the number of parameters, and instantiating a larger model by increasing the model dimensions does not induce overfitting.

## 4.7 Summary of Critical Findings

**Diffuser fails to plan in original state space.** After investigating the performance of diffuser, our findings in Figure 2 indicate that Diffuser fails to successfully replan in high dimensional state spaces when using a Variational Autoencoder (VAE). Based on our observations, we hypothesize that the failure of Diffuser to replan successfully is due to the diffusion objective enabling interpolation

between low density regions in the VAE's latent space by the nature of the training objective smoothing the original probability distribution of trajectories. In practice, this interpolation between low density regions correspond to physically infeasible trajectories. This suggests that interpolation between low density regions in the VAE's latent space is a significant factor in the failure of diffuser to plan and replan successfully. Our results highlight the need for better representation, and for further research of scaling diffusion models to high dimensional control.

**The encoder representation of deep goal-conditioned RL models can be effectively used for hierarchical RL.** We find that having a good representation is essential for effective diffusion in general control settings, as evidenced in Figure 2. Our results imply that the encoder representation of deep reinforcement learning models has the potential to be effectively utilized in hierarchical reinforcement learning. In modern learning from pixels, we propose that an effective task-aware representation can be extracted by using the latent space of an intermediate layer of a low-level policy as an encoder. Our results suggest that using a shared policy encoder between the high and low level policies can improve the effectiveness and efficiency of generative modeling in hierarchical RL.

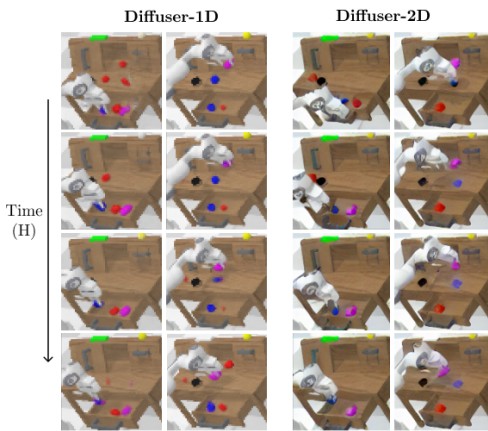

Figure 2: Denoised Latent Representations. Directly using latent diffusion models fails. Hallucination occurs on a $\beta$-TC VAE trained from scratch on the CALVIN dataset (Diffuser-1D), and loss of fine details occurs with SD v1.4's (Rombach et al., 2022) internet-scale pretrained autoencoder (Diffuser-2D). For more and enlarged samples please refer to Appendix K.

## 5 RELATED WORK

**Text-to-Control Models.** Text-to-control models or language-conditioned policies (Lynch & Sermanet, 2020a; Jiang et al., 2019; Colas et al., 2020; Mees et al., 2022a) have been explored in the RL community for improving generalization to novel tasks and environments (Hermann et al., 2017; Bahdanau et al., 2018; Hill et al., 2019; Colas et al., 2020; Wang et al., 2020a). Although they are not the only way to incorporate external knowledge in the form of text to decision making tasks (Luketina et al., 2019; Zhong et al., 2019), they remain one of the most popular (Wang et al., 2020b; Zheng et al., 2022; Duan et al., 2022). However, language grounding in RL remains notoriously difficult, as language vastly underspecifies all possible configurations of a corresponding state (Quine, 1960).

**Diffusion Models.** Diffusion models such as DALL-E 2 (Ramesh et al., 2022) and GLIDE (Nichol et al., 2022) have recently shown promise as generative models, with state-of-the-art text-to-image generation results demonstrating a surprisingly deep understanding of semantic relationships between objects and the high fidelity generation of novel scenes. Video generation models are especially relevant to this work, as they are a direct analog of diffusion planning model Diffuser (Janner et al., 2022) in pixel space without actions. Diffuser first proposed to transform decision making into inpainting and utilize diffusion models to solve this problem, which much work has followed up on (Dai et al., 2023; Chi et al., 2023; Ajay et al., 2022).

For additional related work, please refer to Appendix B.

## 6 CONCLUSION AND LIMITATIONS

Using language to learn atomic sub-skills is critical for complex environments. We use learned state and temporal abstractions to address this, leveraging the unique strengths of diffusion models for long-term planning. Their shortcomings in detailed generation are managed by teaching low-level, goal-driven policies through imitation. Experiments show our model's simplicity and effectiveness, with LCD reaching top performance in a language-based control benchmark. For future work, one could extend LCD to incorporate additional levels of hierarchy and scale to even longer horizons.

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

# Appendix

TABLE OF CONTENTS

Please refer to our website `https://lcd.eddie.win/` for more qualitative results in video format. We release our code and models at `https://github.com/ezhang7423/language-control-diffusion/`.

## A  BROADER IMPACT

In this paper, we present research on diffusion-based models for reinforcement learning. While our work has the potential to advance the field, we recognize the importance of being transparent about the potential societal impact and harmful consequences of our research.

Safety: Our research focuses on the development of diffusion models for reinforcement learning, and we do not anticipate any direct application of our technology that could harm, injure, or kill people.

Discrimination: We recognize the potential for LCD to be used in ways that could discriminate against or negatively impact people. As such, we encourage future work to take steps to mitigate potential negative impact, especially in the areas of healthcare, education, or credit, and legal cases.

Surveillance: LCD did not involve the collection or analysis of bulk surveillance data.

Deception and Harassment: We recognize the potential for our technology to be used for deceptive interactions that could cause harm, such as theft, fraud, or harassment. We encourage researchers to communicate potential risks and take steps to prevent such use of LCD.

Environment: Our research required the usage of GPUs, which may be potentially energy intensive and environmentally costly. However, we seek to minimize the usage of GPUs through LCD's efficiency and the impact of our work could be beneficial to the environment.

Human Rights: We prohibit the usage of LCD that facilitates illegal activity, and we strongly discourage LCD to be used to deny people their rights to privacy, speech, health, liberty, security, legal personhood, or freedom of conscience or religion.

Bias and Fairness: We recognize the potential for biases in the performance of LCD. We encourage researchers building off this work to examine for any biases and take steps to address them, including considering the impact of gender, race, sexuality, or other protected characteristics.

In summary, we recognize the potential societal impact and harmful consequences of our research and encourage researchers to consider these factors when developing and applying new technologies on top of LCD. By doing so, we can help ensure that our work has a positive impact on society while minimizing any potential negative consequences.

## B    ADDITIONAL RELATED WORK

Here we gives a more detailed treatment to additional related work.

### B.1    TEXT-TO-CONTROL MODELS

Modern text to control models often still struggle with long-horizon language commands and misinterpret the language instruction. Hu et al. (2019) attempt to solve a long-horizon Real-Time Strategy (RTS) game with a hierarchical method utilizing language as the communication interface between the high level and low level policy, whilst Harrison et al. (2017) consider training a language encoder-decoder for policy shaping, Hanjie et al. (2021) utilize an attention module conditioned on textual entities for strong generalization and better language grounding. Zhong et al. (2022) propose a model-based objective to deal with sparse reward settings with language descriptions and Mu et al. (2022) also tackle the sparse reward settings through the usage of intrinsic motivation with bonuses added on novel language descriptions. However, only Hu et al. (2019) consider the long-horizon setting, and they do not consider a high-dimensional state space. Jang et al. (2022) carefully examine the importance of diverse and massive data collection in enabling task generalization through language and propose a FiLM conditioned CNN-MLP model (Perez et al., 2018). Much work has attempted the usage of more data and compute for control (Brohan et al., 2022; Jaegle et al., 2021; Reed et al., 2022). However, none of these methods consider the usage of diffusion models as the medium between language and RL.

### B.2    DIFFUSION MODELS

One driver of this recent success in generative modeling is the usage of *classifier-free guidance*, which is amenable to the RL framework through the usage of language as a reward function. The language command is a condition for optimal action generation, which draws direct connection to control as inference (Levine, 2018). Given the success of denoising diffusion probabilistic models (Ho et al., 2020) in text-to-image synthesis (Sohl-Dickstein et al., 2015a), the diffusion model has been further explored in both discrete and continuous data domains, including image and video synthesis (Ho et al., 2022b;c), text generation (Li et al., 2022b), and time series (Rasul et al., 2021). Stable diffusion, an instantiation of latent diffusion (Rombach et al., 2022), has also achieved great success and is somewhat related to our method as they also consider performing the denoising generation in a smaller latent space, albeit with a variational autoencoder (Kingma & Welling, 2013) rather than a low level policy encoder. To expound further on Janner et al. (2022)'s Diffuser, they diffuse the state and actions jointly for imitation learning and goal-conditioned reinforcement learning through constraints specified through classifier guidance, and utilize Diffuser for solving long-horizon and task compositional problems in planning. Instead of predicting the whole trajectory for each state, (Wang et al., 2022b) apply the diffusion model to sample a single action at a time conditioned by state. However, neither of these works considers control from pixels, or utilizing language for generalization. One successful instantatiation of a direct text to video generation model is (Dai et al., 2023). However, they again avoid the issue of directly generating low level actions by using a low-level controller or an inverse dynamics model which we are able to directly solve for.

## C    LOW LEVEL POLICY (LLP) CLARIFICATIONS

We would like to clarify how the state encoder and LLP is trained, how the state encoder is chosen, the specific state encoder that is chosen in HULC, and the reasoning for how we chose HULC as the LLP in our practical instantiation. The state encoder is never trained explicitly, but rather is trained end-to-end as part of the LLP. Thus, after the LLP is trained the state encoder has also been trained. Both the LLP and the state encoder are then frozen during the training of the diffusion model as high-level policy (HLP), as seen in Figure 1.

### C.1    LLP TRAINING OBJECTIVE

As we detail in section 2, in general the loss for the LLP will be some form of behavior-cloning:

$$\mathcal{L}_{\text{BC}} = \mathbb{E}_{s_t, a_t, s_{t+c} \sim \mathcal{D}} \left[ \|a_t - \pi_{\text{lo}}(s_t, g = s_{t+c})\|^2 \right]$$

The specific loss used in HULC for our LLP is described here (Mees et al., 2022a):

$$L_{act}(D_{play}, V) = -\ln(\Sigma_{i=0}^{k} \alpha_k(V_t) \ P(a_t, \mu_i(V_t), \sigma_i(V_t))$$

Here, $P(a_t, \mu_i(V_t), \sigma_i(V_t)) = F(\frac{a_t + 0.5 - \mu_i(V_t)}{\sigma_i(V_t)}) - F(\frac{a_t - 0.5 - \mu_i(V_t)}{\sigma_i(V_t)})$, and $F(\cdot)$ is the logistic distribution's cumulative distribution function.

We choose the state encoder to be the visual encoder within HULC, which has a total of 32 latent dimensions. The state encoding of the current state and some future goal state is then fed to the downstream multimodal transformer and action decoder to generate a new action.

### C.2    REASONING FOR USAGE OF HULC AS LOW LEVEL POLICY (LLP)

We clarify the usage of using a hierarchical model as our LLP here. We would like to clarify that we use a hierarchical approach for our low-level policy for two main reasons. First, we aim to leverage the strongest existing baseline to maximize our chances of creating a SOTA model. Second, we encountered issues replicating the results for the GCBC flat policy from its original paper (Mees et al., 2022b), making it less favorable for our approach. It is also important to note that there is, in principle, no reason why LCD cannot work with a flat policy, and this would be a straightforward addition for future work.

## D    HYPER-PARAMETER SETTINGS AND TRAINING DETAILS

For all methods we proposed in Table 1, Table 3, and Table 6, we obtain the mean and standard deviation of each method across 3 seeds. Each seed contains the individual training process and evaluates the policy for 1000 episodes.

Baselines are run with either 8 Titan RTX or 8 A10 GPUs following the original author guidelines, whilst our experiments are run with a single RTX or A10. In total this project used around 9000 hours of compute.

### D.1    HP AND TRAINING DETAILS FOR METHODS IN TABLE 1.

| Model | Module | Hyperparameter | Value |
|---|---|---|---|
| **HULC** | Trainer | Max Epochs | 30 |
| | | $\beta$ for KL Loss | 0.01 |
| | | $\lambda$ for Contrastive Loss | 3 |
| | | Optimizer | Adam |
| | | Learning Rate | 2e-4 |
| | Model | Transformer Hidden Size | 2048 |
| | | Language Embedding Size | 384 |
| **LCD** | Gaussian Diffusion | Action Dimension | 32 |
| | | Action Weight | 10 |
| | | Loss Type | L2 |
| | | Observation Dimension | 32 |
| | | Diffusion Steps | 20 |
| | | Model Dimension | 64 |
| | Trainer | EMA Decay | 0.995 |
| | | Label Frequency | 200000 |
| | | Sample Frequency | 1000 |
| | | Batch Size | 512 |
| | | Learning Rate | 2e-4 |
| | | Train Steps | 250k |
| | | Number of Steps Per Epoch | 10000 |
| | | Normalizer | Gaussian Normalizer |
| | | Frame Offset | 0 |

Table 6: Hyperparameters for our methods in Table 1.

# E HYPER-PARAMETERS FOR ABLATIONS IN TABLE 5

## E.1 DETAILED INFERENCE PROCEDURE

During inference, we give the MLP a concatenation of the current state $s_i$ and ground truth language, and train the model to output the goal state $s_g = s_{i+c}$, which is then given to the LLP. This is repeated every $c$ time steps. For the transformer, we give the current state $s_i$ and cross-attend to the ground truth language. In order to advantage the ablations, we do not ask the MLP to predict several steps in the future, which been shown to generally fail in practice Chen et al. (2021); Janner et al. (2021; 2022). This simplifies the learning objective and eases the burden on the high-level policy. We keep the same inference procedure for the Transformer in order to ensure fairness. In addition, we give both the MLP and Transformer ablation the ground truth language, so that ablations need only to generalize across goals and not across language.

## E.2 TRANSFORMER HYPERPARAMETERS

We control all other evaluation parameters not listed in Table E.2 to be the same as in our prior experiments for Diffusion high-level policy. These hyperparameters can be found in Appendix D.

| Hyperparameter | Value |
|---|---|
| Number of gradient steps | 100 K |
| Mini-batch size | 512 |
| Transformer hidden dim | 4096 |
| Transformer layers | 4 |
| Final Hidden Activation | ReLU |
| Final Hidden Dim | 4096 |
| Number of heads | 8 |
| Dropout | 0.1 |
| Learning Rate | $2e - 4$ |
| Layer Norm | 32 |
| Total Params | 1.8M |

Table 7: Hyperparameters for the Transformer in Table 5.

We gridsearch over the following parameters and pick the best performing combination from the following:

- Num of transformer layers: [4, 8, 24]
- Width of transformer layers: [2048, 4096, 8192]
- Learning rate: [2e-5, 2e-4, 4e-4]

## F    TRAINING OBJECTIVE DERIVATION

To model this, we turn to diffusion models (Weng, 2021), whom we borrow much of the following derivation from. Inspired by non-equilibrium thermodynamics, the common forms of diffusion models (Sohl-Dickstein et al., 2015b; Ho et al., 2020; Song et al., 2020a) propose modeling the data distribution $p(\boldsymbol{\tau})$ as a random process that steadily adds increasingly varied amounts of Gaussian noise to samples from $p(\boldsymbol{\tau})$ until the distribution converges to the standard normal. We denote the forward process as $f(\boldsymbol{\tau}_t|\boldsymbol{\tau}_{t-1})$, with a sequence of variances $(\beta_0, \beta_1...\beta_T)$. We define $\alpha_t := 1 - \beta_t$ and $\bar{\alpha}_t := \prod_{s=1}^{t} \alpha_s$.

$$f(\boldsymbol{\tau}_{1:T}|\boldsymbol{\tau}_0) = \prod_{t=1}^{T} f(\boldsymbol{\tau}_t|\boldsymbol{\tau}_{t-1}), \quad \text{where } f(\boldsymbol{\tau}_t|\boldsymbol{\tau}_{t-1}) = \mathcal{N}(\boldsymbol{\tau}_t; \sqrt{1-\beta_t}\boldsymbol{\tau}_{t-1}, \beta_t \mathbf{I}). \quad (6)$$

One can tractably reverse this process when conditioned on $\tau_0$, which allows for the construction of a sum of the typical variational lower bounds for learning the backward process' density function (Sohl-Dickstein et al., 2015b). Since the backwards density also follows a Gaussian, it suffices to predict $\mu_\theta$ and $\Sigma_\theta$ which parameterize the backwards distribution:

$$p_\theta\left(\boldsymbol{\tau}_{t-1} \mid \boldsymbol{\tau}_t\right) = \mathcal{N}\left(\boldsymbol{\tau}_{t-1}; \boldsymbol{\mu}_\theta\left(\boldsymbol{\tau}_t, t\right), \boldsymbol{\Sigma}_\theta\left(\boldsymbol{\tau}_t, t\right)\right). \quad (7)$$

In practice, $\Sigma_\theta$ is often fixed to constants, but can also be learned through reparameterization. Following (Ho et al., 2020) we consider learning only $\mu_\theta$, which can be computed just as a function of $\boldsymbol{\tau}_t$ and $\epsilon_\theta(\boldsymbol{\tau}_t, t)$. One can derive that $\boldsymbol{\tau}_t = \sqrt{\bar{\alpha}_t}\boldsymbol{\tau}_0 + \sqrt{1-\bar{\alpha}_t}\epsilon$ for $\epsilon \sim \mathcal{N}(\mathbf{0}, \mathbf{I})$, through a successive reparameterization of (6) until arriving at $f(\boldsymbol{\tau}_t|\boldsymbol{\tau}_0)$. Therefore to sample from $p(\boldsymbol{\tau})$, we need only to learn $\epsilon_\theta$, which is done by regressing to the ground truth $\epsilon$ given by the tractable backwards density. Assuming we have $\epsilon_\theta$, we can then follow a Markov chain of updates that eventually converges to the original data distribution, in a procedure reminiscent of Stochastic Gradient Langevin Dynamics (Welling & Teh, 2011):

$$\boldsymbol{\tau}_{t-1} = \frac{1}{\sqrt{1-\beta_t}}\left(\boldsymbol{\tau}_t - \frac{\beta_t}{\sqrt{1-\bar{\alpha}_t}}\epsilon_\theta\left(\boldsymbol{\tau}_t, t\right)\right) + \sigma_t \mathbf{z}, \quad \text{where } \mathbf{z} \sim \mathcal{N}(\mathbf{0}, \mathbf{I}). \quad (8)$$

To learn $\epsilon_\theta$, we can minimize the following variational lower bound on the negative log-likelihood:

$$\begin{aligned}
L_{\text{CE}} &= -\mathbb{E}_{q(\boldsymbol{\tau}_0)} \log p_\theta(\boldsymbol{\tau}_0) \\
&= -\mathbb{E}_{q(\boldsymbol{\tau}_0)} \log \left( \int p_\theta(\boldsymbol{\tau}_{0:T}) d\boldsymbol{\tau}_{1:T} \right) \\
&= -\mathbb{E}_{q(\boldsymbol{\tau}_0)} \log \left( \int q(\boldsymbol{\tau}_{1:T}|\boldsymbol{\tau}_0) \frac{p_\theta(\boldsymbol{\tau}_{0:T})}{q(\boldsymbol{\tau}_{1:T}|\boldsymbol{\tau}_0)} d\boldsymbol{\tau}_{1:T} \right) \\
&= -\mathbb{E}_{q(\boldsymbol{\tau}_0)} \log \left( \mathbb{E}_{q(\boldsymbol{\tau}_{1:T}|\boldsymbol{\tau}_0)} \frac{p_\theta(\boldsymbol{\tau}_{0:T})}{q(\boldsymbol{\tau}_{1:T}|\boldsymbol{\tau}_0)} \right) \\
&\leq -\mathbb{E}_{q(\boldsymbol{\tau}_{0:T})} \log \frac{p_\theta(\boldsymbol{\tau}_{0:T})}{q(\boldsymbol{\tau}_{1:T}|\boldsymbol{\tau}_0)} \\
&= \mathbb{E}_{q(\boldsymbol{\tau}_{0:T})} \left[ \log \frac{q(\boldsymbol{\tau}_{1:T}|\boldsymbol{\tau}_0)}{p_\theta(\boldsymbol{\tau}_{0:T})} \right] = L_{\text{VLB}}. \\
L_{\text{VLB}} &= L_T + L_{T-1} + \cdots + L_0 \\
\text{where } L_T &= D_{\text{KL}}(q(\boldsymbol{\tau}_T|\boldsymbol{\tau}_0) \parallel p_\theta(\boldsymbol{\tau}_T)) \\
L_t &= D_{\text{KL}}(q(\boldsymbol{\tau}_t|\boldsymbol{\tau}_{t+1}, \boldsymbol{\tau}_0) \parallel p_\theta(\boldsymbol{\tau}_t|\boldsymbol{\tau}_{t+1})) \\
&\quad \text{for } 1 \leq t \leq T-1 \text{ and} \\
L_0 &= -\log p_\theta(\boldsymbol{\tau}_0|\boldsymbol{\tau}_1).
\end{aligned} \quad (9)$$

Which enables us to find a tractable parameterization for training, as the KL between two Gaussians is analytically computable.

$$L_t = \mathbb{E}_{\boldsymbol{\tau}_0, \boldsymbol{\epsilon}} \left[ \frac{1}{2\|\boldsymbol{\Sigma}_\theta(\boldsymbol{\tau}_t, t)\|_2^2} \|\tilde{\boldsymbol{\mu}}_t(\boldsymbol{\tau}_t, \boldsymbol{\tau}_0) - \boldsymbol{\mu}_\theta(\boldsymbol{\tau}_t, t)\|^2 \right]$$

$$= \mathbb{E}_{\boldsymbol{\tau}_0, \boldsymbol{\epsilon}} \left[ \frac{1}{2\|\boldsymbol{\Sigma}_\theta\|_2^2} \| \frac{1}{\sqrt{\alpha_t}} \left( \boldsymbol{\tau}_t - \frac{1-\alpha_t}{\sqrt{1-\bar{\alpha}_t}} \boldsymbol{\epsilon}_t \right) - \frac{1}{\sqrt{\alpha_t}} \left( \boldsymbol{\tau}_t - \frac{1-\alpha_t}{\sqrt{1-\bar{\alpha}_t}} \boldsymbol{\epsilon}_\theta(\boldsymbol{\tau}_t, t) \right) \|^2 \right]$$

$$= \mathbb{E}_{\boldsymbol{\tau}_0, \boldsymbol{\epsilon}} \left[ \frac{(1-\alpha_t)^2}{2\alpha_t(1-\bar{\alpha}_t)\|\boldsymbol{\Sigma}_\theta\|_2^2} \|\boldsymbol{\epsilon}_t - \boldsymbol{\epsilon}_\theta(\boldsymbol{\tau}_t, t)\|^2 \right]$$

$$= \mathbb{E}_{\boldsymbol{\tau}_0, \boldsymbol{\epsilon}} \left[ \frac{(1-\alpha_t)^2}{2\alpha_t(1-\bar{\alpha}_t)\|\boldsymbol{\Sigma}_\theta\|_2^2} \|\boldsymbol{\epsilon}_t - \boldsymbol{\epsilon}_\theta(\sqrt{\bar{\alpha}_t}\boldsymbol{\tau}_0 + \sqrt{1-\bar{\alpha}_t}\boldsymbol{\epsilon}_t, t)\|^2 \right]. \tag{10}$$

After removing the coefficient at the beginning of this objective following (Ho et al., 2020), we arrive at the objective used in the practical algorithm Algorithm 1:

$$\mathbb{E}_{\boldsymbol{\tau}_0, \boldsymbol{\epsilon}}[\|\boldsymbol{\epsilon}_t - \boldsymbol{\epsilon}_\theta(\sqrt{\bar{\alpha}_t}\boldsymbol{\tau}_0 + \sqrt{1-\bar{\alpha}_t}\boldsymbol{\epsilon}_t, t)\|^2]. \tag{11}$$

Furthermore, thanks to the connection between noise conditioned score networks and diffusion models (Song et al., 2020b; Ho et al., 2020), we are able to state that $\epsilon_\theta \propto -\nabla \log p(\boldsymbol{\tau})$:

$$\mathbf{s}_\theta(\boldsymbol{\tau}_t, t) \approx \nabla_{\boldsymbol{\tau}_t} \log p(\boldsymbol{\tau}_t)$$

$$= \mathbb{E}_{q(\boldsymbol{\tau}_0)}[\nabla_{\boldsymbol{\tau}_t} p(\boldsymbol{\tau}_t | \boldsymbol{\tau}_0)]$$

$$= \mathbb{E}_{q(\boldsymbol{\tau}_0)} \left[ -\frac{\boldsymbol{\epsilon}_\theta(\boldsymbol{\tau}_t, t)}{\sqrt{1-\bar{\alpha}_t}} \right] \tag{12}$$

$$= -\frac{\boldsymbol{\epsilon}_\theta(\boldsymbol{\tau}_t, t)}{\sqrt{1-\bar{\alpha}_t}}.$$

Therefore, by using a variant of $\epsilon_\theta$ conditioned on language to denoise our latent plans, we can effectively model $-\nabla_\tau \mathcal{P}_\beta(\boldsymbol{\tau} \mid \mathcal{R})$ with our diffusion model, iteratively guiding our generated trajectory towards the optimal trajectories conditioned on language.

# G   PROOF OF PROPOSITION 3.1

*Proof.* The proof is fairly straightforward, and can be shown by translating our definition of suboptimality into the framework utilized by (Nachum et al., 2019). We are then able to leverage their first theorem bounding suboptimality by the Total Variation (TV) between transition distributions to show our result, as TV is bounded by the Lipschitz constant multiplied by the domain of the function.

(Nachum et al., 2019) first define a low level policy generator $\Psi$ which maps from $S \times \tilde{A}$ to $\Pi$. Using the high level policy to sample a goal $g_t \sim \pi_{\text{hi}}(g|s_t)$, they use $\Psi$ to translate this to a policy $\pi_t = \Psi(s_t, g_t)$, which samples actions $a_{t+k} \sim \pi_t(a|s_{t+k}, k)$ from $k \in [0, c-1]$. The process is repeated from $s_{t+c}$. Furthermore, they define an inverse goal generator $\varphi(s, a)$, which infers the goal $g$ that would cause $\Psi$ to yield an action $\tilde{a} = \Psi(s, g)$. The following can then be shown:

**Theorem G.1.** *If there exists $\varphi : S \times A \to \tilde{A}$ such that,*

$$\sup_{s \in S, a \in A} D_{TV}(P(s'|s, a)||P(s'|s, \Psi(s, \varphi(s, a)))) \leq \epsilon, \tag{13}$$

*then $\text{SubOpt}'(\Psi) \leq C\epsilon$, where $C = \frac{2\gamma}{(1-\gamma)^2} R_{max}$.*

Note that their $\text{SubOpt}'$ is different from ours; whilst we defined in terms of the encoder $\mathcal{E}$ and action generator $\phi$, they define it in terms of $\Psi$. Note, however, that the two are equivalent when the temporal stride $c = 1$, as $\Psi$ becomes $\pi_{\text{lo}} = \phi \circ \mathcal{E}$. It is essential to note that when using a goal conditioned imitation learning objective, as we do in this paper, $\pi_{\text{lo}}$ becomes equivalent to an inverse dynamics model $\text{IDM}(s, \mathcal{E}(s)) = a$ and that $\varphi(s, a)$ becomes equivalent to $\mathcal{E}(s')$. This is the key to our proof, as the second term in the total variation of G.1 reduces to

$$\begin{aligned} &P(s'|s, \Psi(s, \varphi(s, a)))) \\ &= P(s'|s, \Psi(s, \mathcal{E}(s'))) \\ &= P(s'|s, a + \epsilon). \end{aligned} \tag{14}$$

Since we have that the transition dynamics are Lipschitz:

$$\begin{aligned} &\int_{\mathcal{A},\mathcal{S}} |P(s'|s, a) - P(s'|s, a + \epsilon))| \, d\nu \\ &\leq \int_{\mathcal{A},\mathcal{S}} K_f |a - (a + \epsilon)| \, d\nu \\ &= K_f \epsilon \int_{\mathcal{A},\mathcal{S}} d\nu \\ &= K_f \epsilon \, \text{dom}(P(s'|s, a)) \end{aligned} \tag{15}$$

Which we can then plug into 13 to obtain the desired $C = \frac{2\gamma}{(1-\gamma)^2} R_{max} K_f \text{dom}(P(s'|s, a))$.   $\square$

## G.1   CURRENT GAPS BETWEEN THEORY AND PRACTICE

Although we have given a principled motivation for choosing the low level policy state encoder by bounding the suboptimality of our method based on the reconstructed action error, we acknowledge that there still exists several gaps between our theory and practical implementation. Firstly, we assume optimality in our training objective derivation, which may not hold in practice. Specifically in the CALVIN benchmark, the data is suboptimal. Secondly, we do not incorporate the architectural details of the diffusion model into our suboptimality proof, which may yield tighter bounds that what we currently have. Finally, it would be interesting to explicitly analyze the support of the data distribution, in order to give insights into the language generalization capabilities of LCD.

## H  EXPERIMENTAL SETUP DETAILS

### H.1  CALVIN TRAINING DATASET

In this section we give a more detailed account of our training dataset for interested readers.

In order to first train our LLP, we use the original training dataset provided with the CALVN benchmark to perform hindsight relabelling with goal states further down in the trajectory (Andrychowicz et al., 2017). At this point there is no need to incorporate the language labels, as only the high level policy needs to condition its generation on language.

Next, to train our HLP we first collect a new on-policy dataset (as mentioned in Line 1 of Algorithm 1). The training dataset for the high-level policy is generated using trajectories from the low-level policy. Since the original dataset contains language labels for each trajectory, we can condition the low-level policy on a goal state from the original dataset and label this rollout with the corresponding language annotation using this original annotation. We label each rollout with the language embeddings from the 11B T5-XXL model (Raffel et al., 2020), as mentioned in subsection 3.2.

This allows us to effectively tackle a longer horizon control problem than the low-level policy alone, since these generated trajectories are used to conditionally train the high-level policy to generate coherent plans over a longer horizon than the low-level policy. This is achieved by feeding a subsampled trajectory and language embeddings to the high-level policy, and crucially, it generates a sequence of latent goals that leads to the final state. This training objective forces the high level-policy to plan several steps (through predicting a sequence) into the long-term future (through subsampling). In contrast, the low-level policy is only concerned with predicting the very next action given the current state and goal state.

### H.2  FURTHER DETAILS REGARDING CLEVR-ROBOT ENVIRONMENT

Jiang et al. (2019) first introduced the CLEVR-Robot Environment, a benchmark for evaluating task compositionality and long-horizon tasks through object manipulation, with language serving as the mechanism for goal specification. CLEVR-Robot is built on the MuJoCo physics engine, features an agent and five manipulable spheres. Tasked with repositioning a designated sphere relative to another, the agent follows natural language (NL) instructions aligning with one of 80 potential goals. We use 1440 unique NL directives using 18 sentence templates, following Pang et al. (2023). For experimental rigor, we divide these instructions into training and test sets. Specifically, the training set contains 720 distinct NL instructions, corresponding to each goal configuration. Consequently, only these instructions are seen during training. We compare Success Rate (SR) of different methods, with success defined as reaching some L2 threshold of the goal state for a given language instruction. To establish statistical significance, we evaluate 100 trajectories for 3 seeds each. We follow the same training methodology for our method as in the original paper, by first training a Low Level Policy (LLP) on action reconstruction error, and then training a diffusion model to generate goals given language for this LLP.

# I  DIFFUSER-2D

Here we give details for our strongest Diffusion-based ablation, which uses Stable Diffusion's VAE for generating latent plans, which outputs a latent 2D feature map, with height and width 1/8 of the original image. Plans are sampled with a temporal stride of 7, such that each trajectory covers a total of 63 timesteps with $t = 0, 7, 14...63$. Overall, generation quality tends to be higher and more temporally coherent than that of the 1D model, but low level details still not precise enough for control from pixels. For examples of model outputs, please refer to subsection K.2.

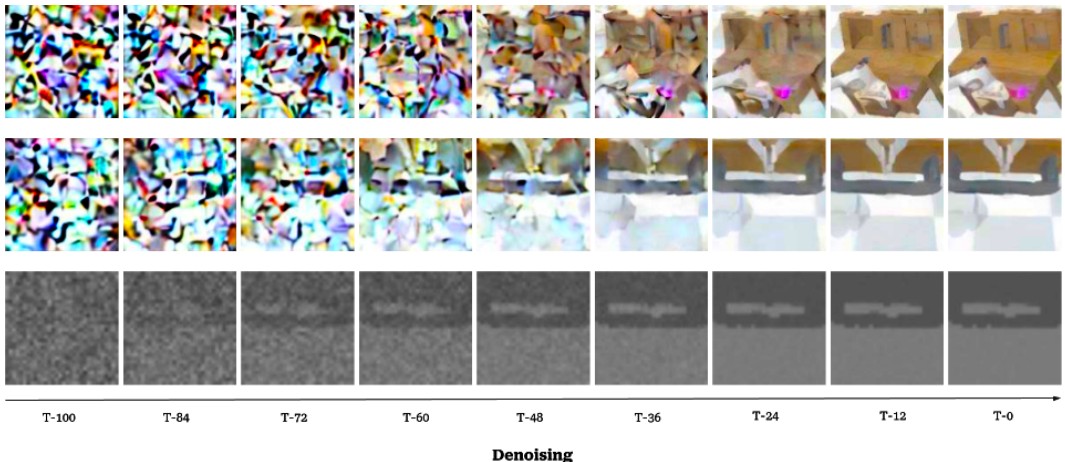

Figure 3: An overview of our Denoising process. In Figure 3 and Figure 4, we give an example of the denoising process of one of our ablations, the Diffuser-2D model. This model utilizes the 2D autoencoder of (Rombach et al., 2022) with (Janner et al., 2022).

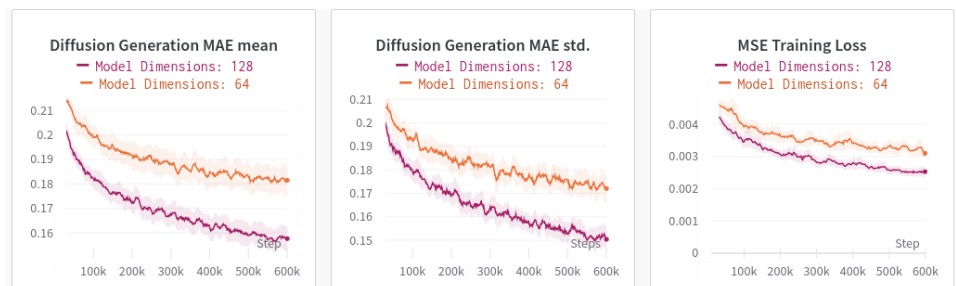

Figure 4: Diffusion Loss Comparison. Here we give study how varying the Diffusion model's size changes the performance of the model. As can be seen, scaling the model from 64 hidden dimensions to 128 strictly increases generation quality, and would likely follow scaling laws observed in (Kaplan et al., 2020).

## J   TASK DISTRIBUTION

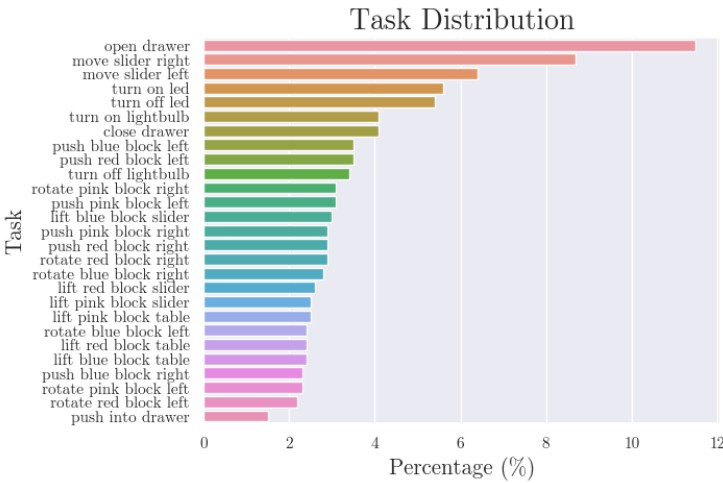

Figure 5: The Evaluation Task Distribution. We visualize the distribution of all the tasks considered in our experiments in Figure 5. Note the long-tailedness of this distribution, and how it skews evaluation scores upwards if one can solve the relatively easier tasks that occur most frequently, such as Open Drawer, Move Slider Right, and Move Slider Left. These tasks only deal with static objects, meaning there is very little generalization that is needed in order to solve these tasks when compared to other block tasks involving randomized block positions.

## K   REPRESENTATION FAILURES

### K.1   DIFFUSER-1D (BETA-TC VAE LATENT REPRESENTATION) FAILURES

We give a few failure cases of decoded latent plans, where the latent space is given by a trained from scratch $\beta$-TC VAE on the CALVIN D-D dataset. The top row of each plan comes from the static camera view, whilst the bottom one comes from the gripper camera view (a camera located at the tool center point of the robot arm). The VAE is trained by concatenating the images in the channel dimension, and compressing to 128 latent dimensions. Plans are sampled with a temporal stride of 9, such that each trajectory covers a total of 63 timesteps with $t = 0, 9, 18...63$. Interestingly, we found that replanning during rollout did not work, precluding the possibility of success on CALVIN with our implementation of this method.

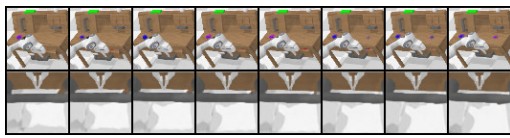

(a) An example of the Close Drawer Task. Notice the flickering block on the top right of the table. Also not the entangled red and blue blocks at the top left of the table.

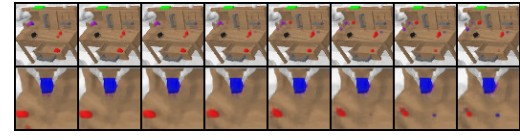

(b) An example of the Lift Blue Block Slider Task. The gripper view is temporally incoherent, red and blue blocks in slider are entangled.

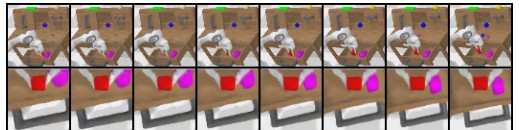

(c) An example of the Lift Red Block Drawer task. Two blocks begin to appear on the table at the end of generation. The red block is also not clearly generated in the first frame.

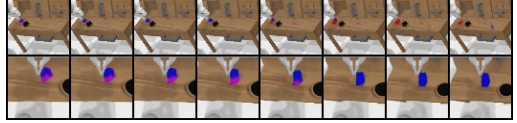

(d) An example of the Push Blue Block Right task. The blue block on the table becomes red by the end of the static view, whereas the opposite happens in the gripper view.

## K.2 DIFFUSER-2D (STABLE DIFFUSION LATENT REPRESENTATION) FAILURES

We additionally give some failure cases for Diffuser-2D. For more information on the training of this model, please refer to Appendix I. We also found that replanning during rollout did not work with this model.

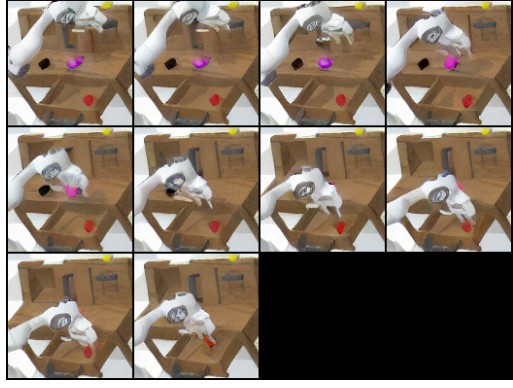

(a) An example of the Lift Red Block Drawer Task. Note the pink block that disappears.

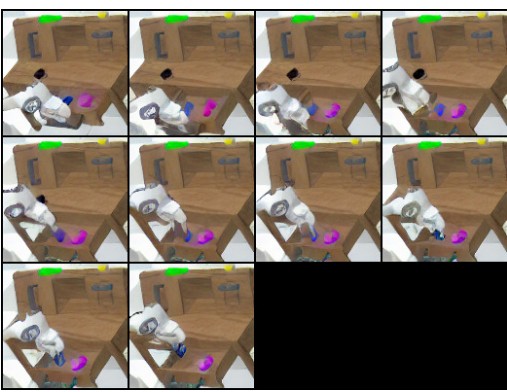

(b) An example of the Lift Blue Block Drawer Task. The gripper arm is entangled with the block.

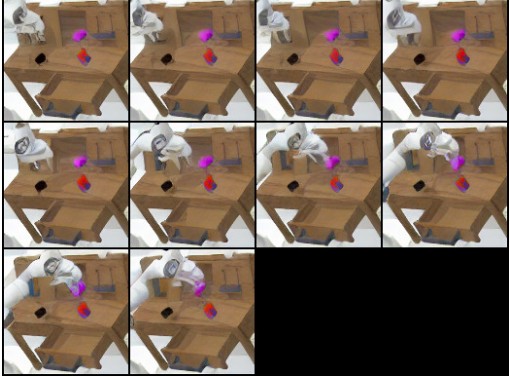

(c) An example of the Lift Pink Block Slider Task. Note the entangled red/blue blocks.

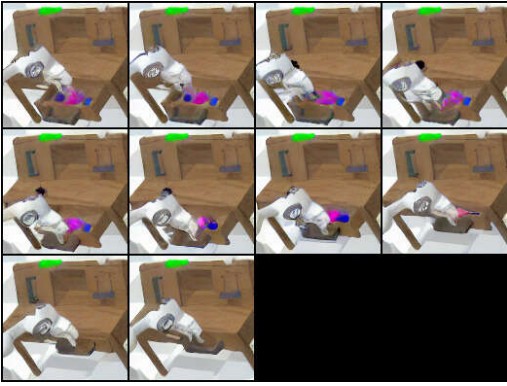

(d) An example of the Close Drawer Task. Note the entangled pink/blue blocks.

## L  TSNE COMPARISON BETWEEN GROUD TRUTH (GT) TRAJECTORY AND DIFFUSER-1D (DM) TRAJECTORY

In order to better understand whether the representation failures found in Appendix K are a result of the underlying encoder or the diffusion model, we visualize the TSNE embeddings of an encoded successful trajectory from the dataset, which we refer to as a Ground Truth trajectory, and the TSNE embeddings of generated trajectories from Diffuser-1D (DM) in Figure 8. If we observe that the DM's embedddings are fairly close to the GT-VAE's, then we can reasonably presume that the VAE is the failure mode, whereas if the trajectories are wildly different this would imply that the DM is failing to model the VAE's latent distribution properly. Here, all samples other than 6 appear to fairly close, so we suspect that the failure case lies in the underlying latent distribution and not the DM's modeling capabilities. This is further backed by LCD, as we show that by using the proper underlying latent space with a LLP leads to success.

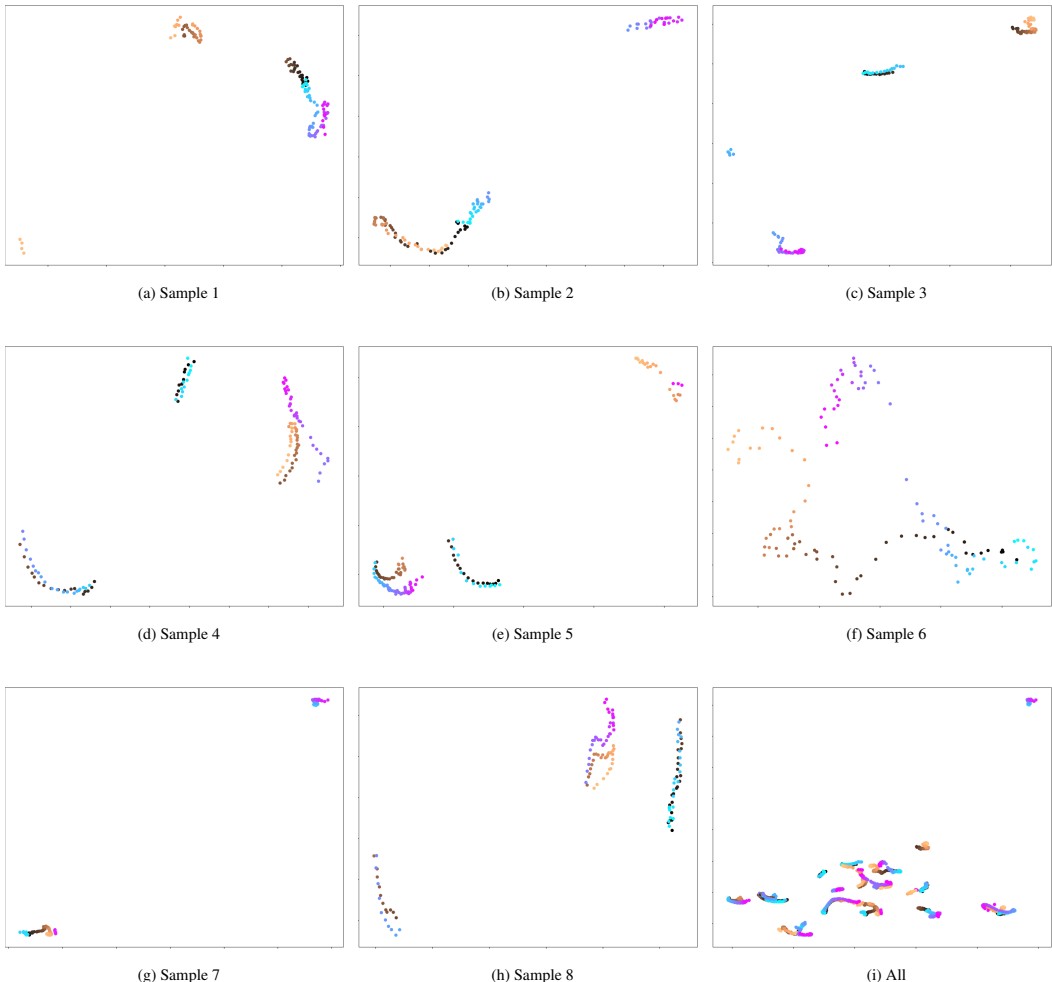

(a) Sample 1          (b) Sample 2          (c) Sample 3

(d) Sample 4          (e) Sample 5          (f) Sample 6

(g) Sample 7          (h) Sample 8          (i) All

Figure 8: TSNE visualization of GT-VAE trajectory vs. Diffuser-1D trajectory, where the purple and light blue color range is the ground truth VAE, and the copper color range is Diffuser-1D. All states are normalized, and all trajectories are taken from the task "lift pink block table".

# M    HULC LATENT PLAN TSNE

We give TSNE embeddings of several Latent Plans generated during inference by HULC below.

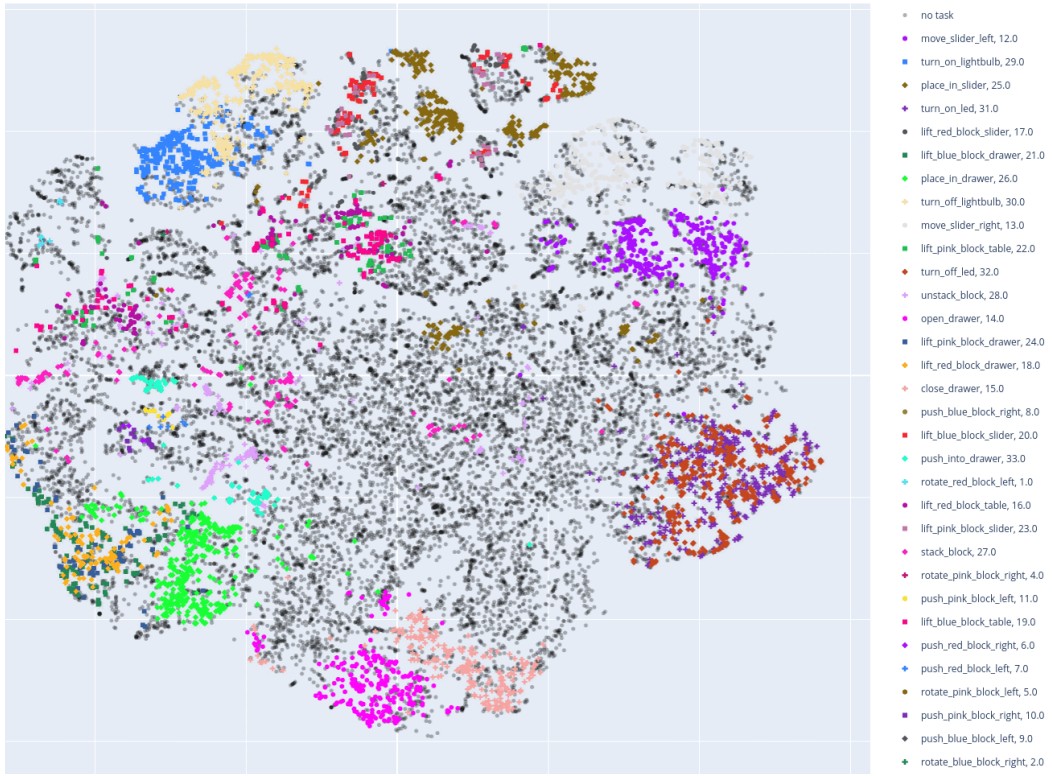

Figure 9: TSNE of Latent Plan. We give a TSNE embedding of the latent plan space of HULC in Figure 9. The latent plan space is the communication layer between the high level policy and low level policy of the HULC model, which corresponds to the intermediate layer between the lower level and lowest level policy in our method. We clarify that this is not the latent goal space that our model does generation in. Our method performs latent generation in the earlier layer from the output of the goal encoder, which corresponds to 32 latent dimensions.

# N    MODEL CARD FOR LANGUAGE CONTROL DIFFUSION

A hierarchical diffusion model for long horizon language conditioned planning.

## N.1    MODEL DETAILS

### N.1.1    MODEL DESCRIPTION

A hierarchical diffusion model for long horizon language conditioned planning.

## N.2    USES

### N.2.1    DIRECT USE

Creating real world robots, controlling agents in video games, solving extended reasoning problems from camera input

### N.2.2    DOWNSTREAM USE

Could be deconstructed so as to extract the high level policy for usage, or built upon further by instantiating a multi-level hierarchical policy

### N.2.3    OUT-OF-SCOPE USE

Discrimination in real-world decision making, military usage

## N.3    BIAS, RISKS, AND LIMITATIONS

Significant research has explored bias and fairness issues with language models (see, e.g., Sheng et al. (2021) and Bender et al. (2021)). Predictions generated by the model may include disturbing and harmful stereotypes across protected classes; identity characteristics; and sensitive, social, and occupational groups.

## N.4    TRAINING DETAILS

### N.4.1    TRAINING DATA

http://calvin.cs.uni-freiburg.de/

## N.5    ENVIRONMENTAL IMPACT

Carbon emissions can be estimated using the Machine Learning Impact calculator presented in Lacoste et al. (2019).

- **Hardware Type:** NVIDIA Titan RTX, NVIDIA A10
- **Hours used:** 9000
- **Cloud Provider:** AWS
- **Compute Region:** us-west-2
- **Carbon Emitted:** 1088.64 kg

## N.6    TECHNICAL SPECIFICATIONS

### N.6.1    MODEL ARCHITECTURE AND OBJECTIVE

Temporal U-Net, Diffusion objective

### N.6.2 COMPUTE INFRASTRUCTURE

**Hardware**   Nvidia Titan RTX , Nvidia A10

**Software**   Pytorch

