# OpenReview forum: "Language Control Diffusion: Efficiently Scaling through Space, Time, and Tasks"
_ICLR.cc/2024/Conference — ICLR 2024 poster_

### Official Review · Reviewer_74cS · 2023-10-30

**Soundness:** 3 good
**Presentation:** 3 good
**Contribution:** 3 good
**Rating:** 6
**Confidence:** 3

**Summary:**

This paper aimed to introduce the hierarchical diffusion policy into robotics control based on language instructions, while this framework faces three challenges: direct long-horizon planning, non-task-specific representation, and computational inefficiency. By proposing a diffusion-based model named Language to Control Diffusion (LCD), this paper addressed these issues.

**Strengths:**

This paper is well-structured and technically sound. The core idea is to pretrain a state encoder and then utilize the latent diffusion model to instantiate high-level policy, generating high-level goals by following the language instruction. The generated goals are then fed into the low-level policy network to obtain the final action.  The empirical results show performance improvements over the previous approaches. The authors also provided many empirical insights of the utilization of diffusion model for visual decision making tasks, which is valuable.

**Weaknesses:**

1. It appears that some crucial descriptions of the model are missing. For instance, there is little mention in the paper about how the state encoder is trained. After going through HULC[Mees et al., 2022a], I just noticed that the encoder is possibly the one in HULC model. Therefore, the author considers both the encoder and LLP to be pretrained. I suggest the author add the corresponding description because it is currently hard to find out this from the logical flow of the paper. It is still important to clarify the training objective and the structure of the state encoder and low-level policy.

2. The novelty of the paper is limited, as it is a combination of diffusion model and HULC.

3. The assumption of the dataset being optimal is very strong in real-world settings, which limits the applicability of the proposed method.

4. The textual encoder -  T5-XXXL model in this paper - is quite large, which increases the inference time. Is it possible to use the textual encoder in CLIP?

**Questions:**

1. It seems low-level policy and the state encoder are coupled in HULC, is it possible to pretrain both modules separately via different objectives?

2. What is the total number of parameters in LCD?

3. Learning state mapping functions and latent planning models separately seems quite reasonable in RL tasks, is there a way to combine a pre-trained encoder on a more extensive visual dataset, similar to models like VQ-VAE used in Stable Diffusion, for state feature extraction, rather than pre-training the encoder solely on robot tasks?

---

> ### Author Response · Authors · 2023-11-15
> **Rebuttal**
>
> > We appreciate the thorough review and the valuable suggestions you've provided regarding the clarity of the state encoder, dataset optimality, and the usage of T5-XXL, each of which has helped us to reflect on and improve the manuscript. We would like to address each of these points specifically.
>
> **Weaknesses:**
>
> 1.  It appears that some crucial descriptions of the model are missing. For instance, there is little mention in the paper about how the state encoder is trained. After going through HULC[Mees et al., 2022a], I just noticed that the encoder is possibly the one in HULC model. Therefore, the author considers both the encoder and LLP to be pretrained. I suggest the author add the corresponding description because it is currently hard to find out this from the logical flow of the paper. It is still important to clarify the training objective and the structure of the state encoder and low-level policy.
>
> >  We have added a [new section in Appendix C](https://i.imgur.com/wvYxnf9.png) in the paper to help clarify the choice of state encoder.
>
> > We confirm that the state encoder is not explicitly trained, but rather is just the output of a hidden layer within the low-level policy. We attempted to illustrate this in [Figure 1](https://i.imgur.com/4OPlRqG.png), although recognize this may have been unclear.
>
> >We have tried our best to clarify our definition and choice of state encoder, and now reference Appendix C in the [introduction](https://i.imgur.com/Ogo0poi.png), the end of the [background](https://i.imgur.com/Ogo0poi.png), and in [Section 3.2](https://i.imgur.com/6TjIN27.png).
>
>
> 2.  The novelty of the paper is limited, as it is a combination of diffusion model and HULC.
>
>  > We respectfully disagree with the reviewer's assessment. It is nontrivial to combine diffusion models as a high level policy (HLP) in hierarchical RL with a low level policy (LLP).
>
> > Naively combining the two immediately leads to several problems - choosing a principled representation for the diffusion model to output becomes an issue, and simply training end-to-end would lead to a nonstationary objective for the diffusion model (as the data distribution is changing) [1]. We have this ablation in [Table 1](https://i.imgur.com/PqpVOGx.png) as Diffuser-1D and Diffuser-2D, which significantly underperform all other models. On the other hand, trying to avoid these issues by training the diffusion model directly to output videos is computationally expensive [2] and also models task-irrelevant details, which may lead to failure in generalization [3] (Please see their [Appendix A.1](https://i.imgur.com/1lDr7uh.png). It is not obvious how to simultaneously solve the nonstationarity, efficiency, and representation problem.
>
> > We propose a novel solution to the representation and efficiency problem by proposing the usage of a frozen pretrained LLP representation. This is not an obvious solution. Furthermore, we give a principled motivation for this choice with our suboptimality theory in Proposition 3.1. Finally, we solve the nonstationarity problem by proposing a latent two-stage training approach, by freezing the LLP during the HLP diffusion training.
>
> > To the best of our knowledge, no other paper has proposed or instantiated a successful latent diffusion planner, let alone with language conditioning. In our work, we are able to demonstrate both the theoretical and empirical effectiveness of such an approach.  Empirically our model significantly outperforms all other methods, and does so 3.3x to 15x faster than other diffusion-based approaches.
>
> >  [1] Sohl-Dickstein et al. Deep Unsupervised Learning using Nonequilibrium Thermodynamics. ICML, 2015.
> > [2] Rombach et al. High resolution image synthesis with latent diffusion models. In  Proceedings of the IEEE/CVF Conference on Computer Vision and Pattern Recognition, pp. 10684–10695, 2022.
> > [3] Du et al. Video Language Planning. arxiv, 2023.
>
> 3.  The assumption of the dataset being optimal is very strong in real-world settings, which limits the applicability of the proposed method.
>
> > Empirically we have found that our method works well even when this optimality assumption is violated, as the data in the CALVIN benchmark is in fact suboptimal [4]. We do acknowledge that there is a gap between the theory and practice and have now explicitly detailed this gap in the paper [here](https://i.imgur.com/bQPcAXt.png)  in Appendix G.
>
> > [4] Mees et al. Calvin: A benchmark for language-conditioned policy learning for long-horizon robot manipulation tasks. IEEE Robotics and Automation Letters, 2022b.
>
> 4.  The textual encoder - T5-XXXL model in this paper - is quite large, which increases the inference time. Is it possible to use the textual encoder in CLIP?
>
> > Thank you for the suggested ablation. We are currently running these experiments now, and will update you when they have finished running.

---

> > ### Author Response · Authors · 2023-11-15
> > **Rebuttal**
> >
> > **Questions:**
> >
> > 1.  It seems low-level policy and the state encoder are coupled in HULC, is it possible to pretrain both modules separately via different objectives?
> >
> >    > Although it is certainly possible, it would require choosing an objective for the state encoder that focuses on modeling task-relevant details and is theoretically justified. One paper that analyzes this can be found here [5], which also concludes that our choice of representation (TARP-BC in their framework) often works well in practice.
> >
> > > [5] Yamada, Jun, et al. "Task-induced representation learning." _arXiv preprint arXiv:2204.11827_ (2022).
> >
> > 2.  What is the total number of parameters in LCD?
> >
> >    > Our high-level policy has 20.1M parameters. Our full method in combination with HULC has 67.8M parameters. For kindly refer you to [Table 3](https://i.imgur.com/AXZZPWT.png) where this information can be found.
> >
> > 3.  Learning state mapping functions and latent planning models separately seems quite reasonable in RL tasks, is there a way to combine a pre-trained encoder on a more extensive visual dataset, similar to models like VQ-VAE used in Stable Diffusion, for state feature extraction, rather than pre-training the encoder solely on robot tasks?
> >
> > > Notably, we actually use this exact VQ-VAE model in Stable Diffusion as a [baseline](https://i.imgur.com/PqpVOGx.png) (Diffuser-2D), where we found that this method does not work well in practice. This is due to the fact that the features extracted by a general pretrained encoder oftentimes do not model information valuable for solving the downstream task, and makes the upstream high level policy learning much more difficult. We also explicitly analyze this in [Section 4.7](https://i.imgur.com/AXZZPWT.png).
> >   ---
> > > We again thank the reviewer for their questions and concerns. In light of these clarifications and sufficient addressing of their concerns, we hope that the reviewer may consider raising their score. We believe that this paper's potential to further scale to longer horizons and improved generalization could make a significant impact on this research topic, and would be extremely grateful for a careful reconsideration of our work.

---

> > > ### Author Response · Authors · 2023-11-16
> > > **Rebuttal - New CLIP Results**
> > >
> > > We would like to let you know that we have finished ablating the choise of T5-XXL by using CLIP, and have attached our results to the main rebuttal. Thank you very much for your questions regarding CLIP. We are looking forward to your feedback.

---

> > > > ### Comment · Reviewer_74cS · 2023-11-20
> > > > **Response**
> > > >
> > > > I want to thank the authors for their explanation and additional experimental results, it does solve some questions for me. I remain somewhat uncertain about whether to accept this paper. The empirical results of this paper, I believe, are quite insightful. However, the novelty of the methodology may be somewhat limited, despite the authors arguing that their method is non-trivial. Therefore, I am unsure about accepting this paper. Nevertheless, considering the inspiration I have drawn from this paper and the authors' earnest responses, I will, for now, raise my score to 6. This does not, however, indicate my final support for the acceptance of this paper. Should any reviewer strongly oppose its acceptance, I will take their opinion into consideration. Conversely, if the majority deems the paper sufficient over the ICLR's bar, I will agree with them.

---

> > > > > ### Author Response · Authors · 2023-11-21
> > > > > **Rebuttal**
> > > > >
> > > > > Dear Reviewer,
> > > > >
> > > > > First and foremost, we would like to express our heartfelt gratitude for your thoughtful feedback and for acknowledging the additional experimental results along with the explanations provided. We understand your concern regarding the perceived novelty of the methodology and appreciate the candidness in expressing your reservations. We believe that our work contributes to the field by not only providing insightful empirical results but also by demonstrating the applicability and utility of utilizing a low level policy representation for hierarchical diffusion.
> > > > >
> > > > > We greatly appreciate you raising your score to 6 in light of the insights you have found in our work and the discussions that have taken place. Your willingness to re-evaluate the paper and adjust your stance based on the responses received is incredibly encouraging for us.
> > > > >
> > > > > It is comforting to know that you have found inspiration from our work, as fostering intellectual stimulation and progression in the field is one of our primary objectives. With respect to the final decision on acceptance, we fully respect and will abide by the collective judgment.

---

### Official Review · Reviewer_FYfw · 2023-10-31

**Soundness:** 3 good
**Presentation:** 3 good
**Contribution:** 2 fair
**Rating:** 6
**Confidence:** 3

**Summary:**

This paper studies text-to-control problems and proposes Language to Control Diffusion models as a hierarchical planner conditioned on language (LCD). This hierarchical approach scales the diffusion model along the spatial, time, and task dimensions.

**Strengths:**

1. The paper is clearly written and easy to follow.
2. The proposed hierarchical approach makes intuitive sense to scale the control diffusion models.
3. The authors provide both theoretical guarantees and experimental results.

**Weaknesses:**

1. The authors stated that the proposed algorithm avoids "the usage of a predefined low level skill oracle". Isn't the low-level controller also adopted by LCD in this work? Do the authors refer to the imitation learning setting where we only access to the trajectories instead of the controller?
2. Can the authors comment more on the difference between the proposed method and previous text-to-video works, such as [1, 2]? Again, the authors mentioned that "they again avoid the issue of directly generating low level actions by using a low-level controller or an inverse dynamics model which we are able to directly solve for", but I didn't see the the downside of leveraging low-level controllers.
3. Besides, I would like to know if these text-to-video methods are directly comparable to LCD in experiments.
4. It is not very clear to me what dom(P(s'|s, a)) measures and the intuitive sense behind it. For terms that are not commonly used in RL theory, it would be better to state its definition and intuition of the proof/bound. It is also not clear what the circle means in the definition of the low-level policy.
5. The theorem comes before the practical instantiation of the algorithm. The gaps between theory and practice should be explicitly analyzed.

[1] Learning Universal Policies via Text-Guided Video Generation.\
[2] VIDEO LANGUAGE PLANNING.

**Questions:**

Please see the weaknesses above.

---

> ### Author Response · Authors · 2023-11-15
> **Rebuttal**
>
> > We appreciate the reviewer's insightful questions and valuable critique. To summarize, we have addressed concerns about the role of low-level controllers, differences between our method and other text-to-video works, the comparability of these methods in experiments, the clarification of certain terms and symbols, and the gaps between theoretical and practical applications in our approach. We trust that our responses below will provide a more comprehensive understanding of our research and its implications.
>
> **Weaknesses:**
>
> 1.  The authors stated that the proposed algorithm avoids "the usage of a predefined low level skill oracle". Isn't the low-level controller also adopted by LCD in this work? Do the authors refer to the imitation learning setting where we only access to the trajectories instead of the controller?
>
> > Here, we refer to two disparate assumptions that we avoid:
> 	> 1. We do not have a perfect low-level controller or inverse dynamics model. This is exactly as you say above. Instead, we must learn a low-level policy (LLP) in our setting.
> 	> 2. We do not define a fixed set of predefined skills between the high-level policy and the low-level controller, also known as the "communication interface".  For example, in Saycan [1] the high level policy chooses between a set of skills like {find apple, place coke, go to table}. In our setting, we do not assume that such a set has been defined.
>
> > [1] Ahn et al. Do as i can, not as i say: Grounding language in robotic affordances, 2022. https://arxiv.org/abs/2204.01691
> 3.  Can the authors comment more on the difference between the proposed method and previous text-to-video works, such as [1, 2]? Again, the authors mentioned that "they again avoid the issue of directly generating low level actions by using a low-level controller or an inverse dynamics model which we are able to directly solve for", but I didn't see the the downside of leveraging low-level controllers.
>
>
> [1] Learning Universal Policies via Text-Guided Video Generation.
> [2] VIDEO LANGUAGE PLANNING.
>
> > Thank you for raising this important point. Here are two reasons why avoiding low-level controllers may be advantageous:
>
> > 1. **Non-Transferability Across Domains**: Many domains do not have a readily available predefined low-level controller, and constructing such a controller requires extensive work that is domain-specific [3]. Once built, these controllers are often not transferable to other contexts, limiting their applications to the scenarios for which they were originally designed. Predefined controllers may restrict the ability of a model to generalize to new contexts or tasks. They are often tailored to specific skill sets and may not support unanticipated actions or sequences that were not considered during their development. This limitation becomes evident when we encounter a requirement for a novel skill or action that the existing low-level controller does not support.
>
> > 2. **Complexity in Construction**: A low-level controller can be intricate to construct [4], particularly in environments that involve many degrees of freedom or require nuanced motor control. The development process can be labor-intensive and may require expert knowledge in robotics, motor control, or specific domain physics.
>
> >[3] Hatze et al. The fundamental problem of myoskeletal inverse dynamics and its implications
>
> >[4] Hitzler et al. "Learning and Adaptation of Inverse Dynamics Models: A Comparison," IEEE-RAS 19th International Conference on Humanoid Robots (Humanoids), Toronto, ON, Canada, 2019, pp. 491-498, doi: 10.1109/Humanoids43949.2019.9035048.
>
> > We defer our answer on the difference between our method and previous text-to-video works below.

---

> ### Author Response · Authors · 2023-11-15
> **Rebuttal**
>
> 4.  Besides, I would like to know if these text-to-video methods are directly comparable to LCD in experiments.
>
> > We give two main points of comparison between text-to-video methods and LCD here:
>
> > 1. **Robustness of Communication Interface**: Text-to-video methods rely heavily on a high-quality decoder or skill oracle capable of generalizing well across various contexts  (see [Appendix A.1](https://i.imgur.com/1lDr7uh.png) of Video Language Planning). The quality of these components is critical for their success, and performance can degrade if these elements do not generalize well. Our approach incorporates this inductive bias and performs diffusion directly in the latent goal space of the decoder rather than in the image space, which often hallucinates (see [Figure XII of Video Language Planning](https://i.imgur.com/VvI9Wvx.png),  [related work in Tune-A-Video](https://i.imgur.com/S0mWtGe.png) [5]) and is significantly more computationally expensive.  Thus, our method potentially offers more consistent results in a wider range of scenarios.
>
> > 2. **Efficiency and Accessibility**: Our LCD method was designed with efficiency in mind. Text-to-video methods may indeed give better performance in some scenarios where large-scale compute is accessible. However, this is often not an option for individual researchers or small-scale labs. LCD provides an agile and less resource-demanding alternative, enabling rapid iteration and experimentation with diffusion planners.
>
> > In conclusion, when comparing with text-to-video methods, LCD offers a valuable and pragmatic solution for research settings where computational resources or the ability to develop complex decoders and communication interfaces are limited. We believe that the LCD's advantages, particularly in terms of efficiency and robustness, make it a worthwhile alternative for many practical applications.
>
> > [5] Wu, Jay Zhangjie, et al. "Tune-a-video: One-shot tuning of image diffusion models for text-to-video generation." _Proceedings of the IEEE/CVF International Conference on Computer Vision_. 2023.
>
> 5.  It is not very clear to me what dom(P(s'|s, a)) measures and the intuitive sense behind it. For terms that are not commonly used in RL theory, it would be better to state its definition and intuition of the proof/bound. It is also not clear what the circle means in the definition of the low-level policy.
>
> > $\text{dom}(P(s'|s, a))$ is the size of the set of possible next states, or more formally the cardinality of the domain of the transition dynamics function. In the definition $\pi_{lo}(s) := \phi \circ E(s)$  ,  the $\circ$ symbol refers to function composition. We have added these definitions for clarity [in Section 3.1](https://i.imgur.com/xcu1Pik.png).
>
> 7.  The theorem comes before the practical instantiation of the algorithm. The gaps between theory and practice should be explicitly analyzed.
>
> > We appreciate the reviewer's thoughtful recommendation. Following this guidance, we have elaborated on this aspect and included a new section in [Appendix G](https://i.imgur.com/bQPcAXt.png) of our paper. We trust this addition clarifies the point and strengthens our argument.
>
> > We believe that our comprehensive rebuttal and modifications emphasize the strength and scalability of our research, with the capacity to influence future advancements in the field. We hope that our responses have sufficiently met your queries and concerns. This paper's potential to enhance scalability and advancement toward longer horizons could greatly contribute to this research area. Thank you very much for your thoughtful evaluation of our research, and we eagerly look forward to your continued feedback.

---

> > ### Comment · Reviewer_FYfw · 2023-11-22
> >
> > This addresses most of my concerns. Thank the authors for the clarification.

---

### Official Review · Reviewer_z6e9 · 2023-11-03

**Soundness:** 3 good
**Presentation:** 3 good
**Contribution:** 2 fair
**Rating:** 5
**Confidence:** 4

**Summary:**

In this work, the authors propose to scale diffusion models for planning in extended temporal, state, and task dimensions to tackle long-horizon control problems conditioned on natural language instructions. Specifically, they take a hierarchical diffusion approach by training diffusion policy to plan in the latent plan (induced by a frozen low-level policy) every c steps (temporal abstraction). They leverage the language-conditioning capabilities of existing diffusion architecture to learn language-conditioned hierarchical policies.

**Strengths:**

+ The proposed approach achieves SOTA performance on a recently proposed language robotics benchmark
+ The method is well-motivated and reasonable

**Weaknesses:**

The weakness comes from a combination of lack of originality and broadness of the experiments. While the approach is very reasonable (applying language-conditioned latent diffusion in the abstracted state and action space for high-level planning), its general idea is similar to the ones in the literature (e.g., [1]). The major differences are the use of diffusion models and the specific choice of temporal abstraction. Based on this, I would like to see more experiment evidence, e.g., beyond the CALVIN benchmark, or real-world experiments (as in SPIL).

[1] Parrot: Data-Driven Behavioral Priors for Reinforcement Learning

**Questions:**

See “weaknesses”

---

> ### Author Response · Authors · 2023-11-15
> **Rebuttal**
>
> > We are grateful to the reviewer for their valuable comments, which allows us to emphasize the novelty of our diffusion model framework. We also justify our choice of the CALVIN benchmark as a testing ground and respond to requests for more extensive experimental evidence.
>
> **Weaknesses:**
>
> The weakness comes from a combination of lack of originality and broadness of the experiments. While the approach is very reasonable (applying language-conditioned latent diffusion in the abstracted state and action space for high-level planning), its general idea is similar to the ones in the literature (e.g., [1]). The major differences are the use of diffusion models and the specific choice of temporal abstraction.
>
> > We respectfully disagree with the reviewer's assessment. We would like to clarify that our main novelty comes from being the first to propose utilizing the implicit encoder of a pretrained low level policy for use in a latent language diffusion model and demonstrating both the effectiveness and efficiency of such an approach. On the other hand, PARROT [1] does not incorporate language and therefore must learn a new high level policy per task. In addition, it uses a generative model (normalizing flow) for the low-level policy (LLP), which is a more restrictive class of models than general low level policies. We are able to generalize [zero-shot to new tasks](https://i.imgur.com/zniB6aD.png) through language, and do not need to make such assumptions on the LLP.
>
> > Our method innovatively integrates diffusion models into hierarchical RL as a high-level policy, overcoming common pitfalls. A naive approach combining the two grapples with an unprincipled nonstationary objective [2] if trained end-to-end, which we show in [Table 1](https://i.imgur.com/PqpVOGx.png) as Diffuser-1D and Diffuser-2D. These baselines significantly underperform all other models. Alternative strategies, like two-stage training a text-to-video diffusion model after training the LLP, are not only resource-intensive but also capture unnecessary details that impede generalization [3, 4].
>
> > Our approach crafts a novel solution to these challenges by leveraging a frozen, pretrained low-level policy representation – a non-obvious choice that we defend with our suboptimality theory (Proposition 3.1). We further address the nonstationarity issue with a latent two-stage training protocol, enhancing both efficiency and performance. Empirically our model significantly outperforms all other methods, and does so 3.3x to 15x faster than other diffusion-based approaches.
>
> > To the best of our knowledge, no other paper has proposed or instantiated a successful latent diffusion planner, let alone with language conditioning. In our work, we are able to demonstrate both the theoretical and empirical effectiveness of such an approach.
>
> >  [1] Parrot: Data-Driven Behavioral Priors for Reinforcement Learning
> >  [2] Sohl-Dickstein et al. Deep Unsupervised Learning using Nonequilibrium Thermodynamics. ICML, 2015.
> > [3] Rombach et al. High resolution image synthesis with latent diffusion models. In  Proceedings of the IEEE/CVF Conference on Computer Vision and Pattern Recognition, pp. 10684–10695, 2022.
> > [4] Du et al. Video Language Planning. arxiv, 2023. (Please see their [Appendix A.1](https://i.imgur.com/1lDr7uh.png))
>
> Based on this, I would like to see more experiment evidence, e.g., beyond the CALVIN benchmark, or real-world experiments (as in SPIL).
>
> > Our work stands out as the first to leverage a pretrained latent encoding within a new diffusion planning framework, offering unique efficiency and effectiveness as substantiated by our results.
>
> > We would like to emphasize that we believe the current experimental evaluation is sufficient for showing the effectiveness of our LCD approach. The choice of the CALVIN benchmark was deliberate; it is a well-established and challenging benchmark that tests for both the efficacy and efficiency of high-level planning and control in a complex environment. Our results on this benchmark clearly demonstrate the practical benefits of the proposed LCD method over existing techniques, thereby validating our contributions despite the perceived overlap with earlier works.
>
> > Moreover, we are also aware of the imperative to demonstrate the generalizability and applicability of LCD. To this end, we are in the process of setting up additional experiments on the CLEVR Robot Env and will update you later with these findings. These will further substantiate our claims and illustrate the versatility and robustness of the LCD framework beyond the CALVIN benchmark.
>
> > In conclusion, while we acknowledge the reviewer’s call for further experimental evidence, we maintain that our current experiments are both relevant and convincing for the scope of this study. Additional experiments, though underway, should be seen as extending rather than essential to the core contributions of the work.

---

> > ### Author Response · Authors · 2023-11-15
> > **Rebuttal**
> >
> > > We are grateful to you for your insightful queries and comments. We hope our clarification and comprehensive responses adequately address the concerns raised and encourage a review of the score assigned. We are confident that the scalability and improved generalization of our paper could have significant impact on this field of research. We kindly request a reevaluation of our work, considering its potential contribution to this area, and thank you for your time and consideration.

---

> > > ### Author Response · Authors · 2023-11-21
> > > **Rebuttal**
> > >
> > > We are pleased to inform you that other reviewers have updated their assessments, and the ratings now indicate an acceptance for our paper. We would like to extend our gratitude for the insights and feedback that you have provided during the review process. Should you have any additional questions or concerns, we are available to address them promptly. However, if there are no further inquiries, we kindly ask you to reconsider the update of your rating to acceptance, aligning with the consensus of the other reviewers. Your understanding and collaboration are highly valued, and we believe that aligning the reviews would reflect a cohesive evaluation of our work.

---

> ### Author Response · Authors · 2023-11-23
> **New benchmark results**
>
> Dear z6e9,
>
> We wish to bring to your attention that we have posted new benchmark results, providing new data that may have a bearing on the evaluation of our work. These results have been rigorously obtained and we believe they contribute positively to the overall strength and validity of our research.
>
> We trust that the supplementary results and thorough answers provided sufficiently address the issues highlighted, prompting a reevaluation of the initially given score. Thank you very much for your time and thoughtful consideration. We are grateful for the expertise and effort you bring to the review process.

---

### Official Review · Reviewer_fAWH · 2023-11-03

**Soundness:** 3 good
**Presentation:** 3 good
**Contribution:** 3 good
**Rating:** 6
**Confidence:** 3

**Summary:**

The authors propose a novel hierarchical framework called LCD, which leverages a language-conditioned diffusion model as a high-level goal planner on top of HULC. The proposed model achieves scalability in the spatial, time, and task dimensions. It outperforms other baselines on the CALVIN benchmark.

**Strengths:**

1. The proposed method is simple and effective, achieving scalability in the spatial, time, and task dimensions.
2. By employing a hierarchical decision-making approach, the algorithm reduces the model size and inference time.
3. The authors experimentally demonstrate that vanilla Diffuser fails to successfully replan in high-dimensional state spaces when using a Variational Autoencoder (VAE). This provides insights for researchers interested in using Diffuser for planning tasks with image state spaces.

**Weaknesses:**

1. In Section 4.5 of the experiments, the parameter sizes of MLP and Transformer are significantly smaller than the parameters of Diffusion. This may introduce unfairness in the experiments. Additionally, I did not notice a detailed explanation of how the authors perform inference using MLP and Transformer. Since the authors employ a Diffusion planning paradigm as HLP, it may be worth considering a better comparison with planning-based MLP (e.g., model-based RL) and Transformer (e.g., Decision Transformer).

**Questions:**

1. The experimental results seem to indicate that HULC plays an important role in the success of LCD. Do the authors believe that the performance of LCD necessarily relies on a well-trained HULC baseline? Can LCD still achieve excellent performance with a simple goal-conditioned policy and a well-designed RL encoder representation? Alternatively, if a HULC baseline is used, can a Transformer with a parameter size comparable to the Diffusion HLP achieve similar performance?

---

> ### Author Response · Authors · 2023-11-15
> **Rebuttal**
>
> **fAWH**
>
> > We thank the reviewer for their perceptive questions and feedback on the performance of inference using the MLP and Transformer ablation, considerations of parameter size in making comparisons, and the role of HULC as a baseline in LCD. In the responses following, we have sought to address each point in detail. We hope our clarifications will elucidate a better understanding of our approach and its potential to influence this research field.
> >
> **Weaknesses:**
>
> 1.  Additionally, I did not notice a detailed explanation of how the authors perform inference using MLP and Transformer.
>
> > Regarding the inference procedure using MLP and Transformer, we have since added a [new section](https://i.imgur.com/MwzAO6s.png) in Appendix E that provides a detailed explanation. Appendix E also includes a more detailed treatment of the hyperparameters for the transformer ablation.
>
>  In Section 4.5 of the experiments, the parameter sizes of MLP and Transformer are significantly smaller than the parameters of Diffusion. This may introduce unfairness in the experiments.
> > We would like to note that for this experiment we rigorously tested several model sizes for the Transformer to ensure a fair comparison, ultimately reporting results for the most performant Transformer; this Transformer has a smaller parameter count than our Diffusion model. The hyperparameters for this gridsearch can be found in [Appendix E.2](https://i.imgur.com/bBuJJPf.png). However, we found the performance of the Transformer is still significantly lower than LCD, as evidenced in Section 4.5.   Notably, we have since run [further experiments](https://i.imgur.com/b4vmUgj.png) with a similar-sized MLP (25.5M params vs LCD 20.1M params) that further reinforce our original findings.
>
>  Since the authors employ a Diffusion planning paradigm as HLP, it may be worth considering a better comparison with planning-based MLP (e.g., model-based RL) and Transformer (e.g., Decision Transformer).
>
>  > In this setting, we strove to give the ablations as much advantage as possible, and in order to keep simplicity and ease the learning objective for the ablations we did next-state prediction rather than planning. Generally, it has been found empirically that the MLP does not work well in MBRL and trajectory generation [1, 2], so we chose only next-state prediction as the objective. We gave the same objective for the transformer to ensure uniformity.
>
>  >[1] Janner, et al. Offline reinforcement learning as one big sequence modeling problem,  Advances in neural information processing systems 2021.
>
> > [2] Chen, et al. Decision transformer: Reinforcement learning via sequence  modeling.  Advances in neural information processing systems, 2021
> >
> **Questions:**
>
> 3.  The experimental results seem to indicate that HULC plays an important role in the success of LCD. Do the authors believe that the performance of LCD necessarily relies on a well-trained HULC baseline? Can LCD still achieve excellent performance with a simple goal-conditioned policy and a well-designed RL encoder representation? Alternatively, if a HULC baseline is used, can a Transformer with a parameter size comparable to the Diffusion HLP achieve similar performance?
>
> >  We would like to clarify that we used HULC for our low-level policy for two main reasons. First, we aimed to leverage the strongest existing baseline to maximize our chances of creating a SOTA model. Second, we encountered issues replicating the results for the GCBC flat policy from its original paper, making it less favorable for our approach. It is important to note that there is, in principle, no reason why LCD cannot work with a flat policy, and this would be a straightforward addition for future work. We kindly refer you to [Appendix C](https://i.imgur.com/Fl1mYy2.png), where we have also explicitly clarified this point in our paper.
>
> > We want to reiterate that we did not find a Transformer with a comparable parameter size that was able to achieve similar performance to LCD.
>
> ---
> > We are confident that our thorough response and amendments underscore the robustness and scalability of our work, potentially driving future progress in the field. We trust that these responses adequately address your concerns. We believe that this paper's potential to further scale to longer horizons and improved generalization could make a significant impact on this research topic, and are extremely grateful for your review of our work. We look forward to your continued feedback.

---

### Author Response · Authors · 2023-11-15
**Rebuttal Summary**

We extend our sincere gratitude to all the reviewers for their diligent evaluations and constructive feedback. Below, we summarize the new experiments that we have run and key modifications we have made to the manuscript based on the reviewers' input.

## New Experiments

- We have run our MLP (25.5M params) as high level policy (HLP) ablation with a comparable parameter count as LCD (20.1M params), and [found that](https://i.imgur.com/OAcFMQv.png) the comparable-sized MLP ablation still significantly underperforms our model.
* We are ablating the T5-XXL choice of encoder now by running LCD with CLIP embeddings, and will update later with results.
*  We are also running LCD on the CLEVR robot env in order to extend our empirical evaluation to more benchmarks.

## Modifications to the paper

A list of modifications to our manuscript are given below:

- Added a new section in Appendix C providing clarity on the state encoder's role and integration within the low-level policy. Updated the introduction, background, and Section 3.2 to point towards it.
- Incorporated additional discussions about the gaps between theory and practice, specifically on the success of our method with suboptimal data and other practical implications in Appendix G.
- Addressed concerns in our ablation experiment regarding parameter size and comparison fairness by conducting additional experiments, results of which have been incorporated into Section 4.5.
- Introduced a more thorough explanation of technical terms like `dom(P(s'|s, a))` and symbols such as the function composition circle in Section 3.1.
 - Added a detailed treatment of the inference procedure for ablations in Section 4.5 to Appendix E.


In response to the constructive criticism received, we trust these revisions bolster the presentation of our research. We believe that the changes made aptly demonstrate the robustness, scalability, and innovative nature of our work. We humbly request the reviewers to reassess our manuscript considering our response, and believe that this further clarification helps position our paper as a meaningful contribution to the field. We look forward to any additional suggestions that will benefit our work and thank the reviewers once again for their continued feedback.

---

> ### Author Response · Authors · 2023-11-16
> **Rebuttal - New Results**
>
> # CLIP Ablation Results
>
> We have ablated the T5-XXL choice of encoder now by running LCD with CLIP embeddings, and include the results below. We use the same training methodology as in our original paper for Table 1, and rigorously follow the CALVIN benchmark’s evaluation procedure by running 3 seeds with our method, each seed evaluating for 1000 episodes. Numbers for HULC with CLIP are taken from the original author's paper [1].
>
> ---
> | Horizon    | HULC            | Ours  (LCD)               |
> |------------|-----------------|--------------------|
> | One        | 81.4 ± 0.5      | **84.9 ± 0.96**    |
> | Two        | 60.4 ± 1.1      | **64 ± 2.62**      |
> | Three      | 44.7 ± 2.3      | **48.77 ± 4.39**   |
> | Four       | 32.3 ± 1.2      | **36.17 ± 5.13**   |
> | Five       | 23.2 ± 1.6      | **25.93 ± 4.94**   |
> | Avg Horizon Length | 2.42 ± 0.06      | **2.60 ± 0.17**    |
>
> Table 1: Comparison of HULC and LCD performance metrics
>
> ---
>
> Across all horizons, our method (LCD) significantly outperforms HULC, as indicated by the heavier weighting in bold. In addition, we find that the CLIP embeddings are not as performant as the T5-XXL embeddings, indicating that CLIP potentially leads to worse generalization across language for robotic tasks than T5-XXL.
>
> [1] Mees et al. What matters in language conditioned robotic imitation learning over unstructured data, 2022a. URL https://arxiv.org/abs/2204.06252.

---

> > ### Author Response · Authors · 2023-11-23
> > **Rebuttal - New Benchmark Results**
> >
> > # CLEVR-Robot Env Results
> >
> > We are pleased to report to the reviewers that, during the rebuttal process, we have obtained significant new results for the CLEVR-Robot Environment.
> >
> >
> > **Dataset and Metric**
> >
> > In order to show the general effectiveness of our approach, we have rigorously run new experiments on the CLEVR-Robot Environment. Jiang et al. [1] first introduced the CLEVR-Robot Environment, a benchmark for evaluating task compositionality and long-horizon tasks through object manipulation, with language serving as the mechanism for goal specification.  CLEVR-Robot is built on the MuJoCo physics engine, features an agent and five manipulable spheres. Tasked with repositioning a designated sphere relative to another, the agent follows natural language (NL) instructions aligning with one of 80 potential goals. We use 1440 unique NL directives using 18 sentence templates. For experimental rigor, we divide these instructions into training and test sets. Specifically, the training set contains 720 distinct NL instructions, corresponding to each goal configuration. Consequently, only these instructions are seen during training. We compare Success Rate (SR) of different methods, with success defined as reaching some L2 threshold of the goal state for a given language instruction. To establish statistical significance, we evaluate 100 trajectories for 3 seeds each. We follow the same training methodology for our method as in the original paper, by first training a Low Level Policy (LLP) on action reconstruction error, and then training a diffusion model to generate goals given language for this LLP.
> >
> > **Baselines for comparison**
> >
> > We consider the following representative baselines: a Goal-Conditioned Behavior Cloning Transformer, which is trained end to end by predicting the next action given the current observation and language instruction. We use a comparable parameter count  for each model to ensure fairness. This establishes the most straightforward approach to this benchmark. We also consider using a Hierarchical Transformer, which is trained using the same two-stage training method as our method. The Hierarchical Transformer is equivalent to LCD (our method), other than the usage of a transformer rather than a diffusion model for the high-level policy.
> >
> >
> > **Table 1**
> >
> > |        |  Hierarchical Transformer  | GCBC | Ours       |
> > |--------|------------|--------------------------|------------|
> > | Seen   | 38.0 ± 2.2 | 45.7 ± 2.5               | **53.7 ± 1.7** |
> > | Unseen |  36.7 ± 1.7 | 46.0 ± 2.9                | **54.0 ± 5.3** |
> > | Noisy  | 33.3 ± 1.2     |42.7 ± 1.7            | **48.0 ± 4.5** |
> >
> >
> > The results presented in Table 1 demonstrate the efficacy of LCD. Across all settings—Seen, Unseen, and Noisy—LCD outperforms both the Goal-Conditioned Behavior Cloning (GCBC) Transformer and the Hierarchical Transformer .
> >
> > The "Seen" setting refers to the agent's performance on Natural Language (NL) instructions that were included in the training set. LCD's superior performance in this context with **53.7 ± 1.7** SR suggests that it can effectively learn and recreate the tasks it has been trained on.
> >
> > The "Unseen" setting evaluates the model's generalization abilities to novel tasks, by evaluating on NL instructions that were not encountered during training. Again, LCD reports the highest SR with **54.0 ± 5.3**, indicating its robustness and ability to understand and execute language instructions beyond its training scope.
> >
> > The "Noisy" setting simulates real-world conditions where there might be noise in the agent's language input. We add noise to each NL instruction, by dropping and inserting random words.  LCD maintains a lead in this more challenging scenario with an SR of **48.0 ± 4.5**. Despite a drop in performance compared to the other settings, this result showcases the robustness of the diffusion model to errors in language.
> >
> >
> > ----
> > We have revised our paper to include these [new results](https://imgur.com/a/QIVGeDS). We trust that the new evaluations, demonstrating LCD's superior performance across various conditions, will help address any concerns the reviewers may have and further show the general effectiveness of our method. We appreciate the careful consideration that the reviewers have already dedicated to evaluating our paper, and we look forward to any further insights they may offer.
> >
> > [1] Yiding Jiang, Shixiang Gu, Kevin Murphy, and Chelsea Finn. Language as an Abstraction for
> > Hierarchical Deep Reinforcement Learning. Curran Associates Inc., Red Hook, NY, USA, 2019.

---

### Meta-Review · Area_Chair_vhit · 2023-12-11

**Metareview:**

The paper addresses the problem of language conditioned policy learning. It builds upon HULC, a Hierarchical Language-Conditioned Imitation Learning approach for robotic control, and proposes to use a diffusion model as a high level planner. In particular, a text-to-latent plan diffusion model is trained. The latent plan is the condition to the low level policy.

All reviewers find the proposed approach intuitive and timely -- utilizing the power of diffusion models to condition on language and predict over longer time horizons; operating in latent space allows for a smaller and more efficient model.

Further, the reviewers appreciate the strength of the results on CALVIN and presented analysis, e.g. speed evaluation and justification of use of diffusion.

On the other hand, two of the reviewers express concerns with the novelty of the approach, in particular this being a simple extension of HULC. Further, they ask for more experiments to justify the proposed approach. In addition, there is confusion regarding some parts of the approach, e.g. how the low level policy and the state encoders are being trained and integrated.

**Justification For Why Not Higher Score:**

Several reviewers are concerned that the approach is a straightforward extension of the HULC. The authors' response of challenges with a direct integration of Diffusion Models within HULC being nontrivial are not convincing to at least one reviewer. Further, the reviewers find some parts not well explained -- how are the low level policy and state encoder trained and integrated.

**Justification For Why Not Lower Score:**

The paper received 1 x borderline reject and 3 x borderline accept.

The reviewers appreciate the use of diffusion and language models in a hierarchical setup. In particular, the use of diffusion models as latents and their ability to be language conditioned allows for multi-task policies that one can naturally interface with. The approach is substantiated by strong results on CALVIN, a standard and challenging benchmark for the problem space, as well as on CLEVR-Env.

The only borderline reject rating is justified by lack of novelty. We believe the rebuttal addresses these concerns by showing strong results on an additional benchmark and clarifying the intricacies of using diffusion models and the differences to HULC.

---

### Decision · Program_Chairs · 2024-01-16

Accept (poster)